# SKADA-Bench: Benchmarking Unsupervised Domain Adaptation Methods with Realistic Validation On Diverse Modalities

**Yanis Lalou**[*]                                              *yanis.lalou@polytechnique.edu*
*École Polytechnique, IP Paris, CMAP, UMR 7641, 91120 Palaiseau, France*

**Théo Gnassounou**[*]                                          *theo.gnassounou@inria.fr*
*Université Paris-Saclay, Inria, CEA, 91120 Palaiseau, France*

**Antoine Collas**[*]                                           *antoine.collas@inria.fr*
*Université Paris-Saclay, Inria, CEA, 91120 Palaiseau, France*

**Antoine de Mathelin**[*]                                      *antoine.demat@gmail.com*
*Centre Borelli, ENS Paris-Saclay, Gif-sur-Yvette, 91190, France*

**Oleksii Kachaiev**                                            *oleksii.kachaiev@gmail.com*
*MaLGa Center - Dipartimento di Matematica - Università degli Studi di Genova, Italy*

**Ambroise Odonnat**                                            *ambroise.odonnat@gmail.com*
*Inria, Univ. Rennes 2, CNRS, IRISA, Paris, France*

**Thomas Moreau**                                               *thomas.moreau@inria.fr*
*Université Paris-Saclay, Inria, CEA, 91120 Palaiseau, France*

**Alexandre Gramfort**                                          *alexandre.gramfort@inria.fr*
*Université Paris-Saclay, Inria, CEA, 91120 Palaiseau, France*

**Rémi Flamary**                                                *remi.flamary@polytechnique.edu*
*École Polytechnique, IP Paris, CMAP, UMR 7641, 91120 Palaiseau, France*

**Reviewed on OpenReview:** *https://openreview.net/forum?id=k9F63DV3Qe*

## Abstract

Unsupervised Domain Adaptation (DA) consists of adapting a model trained on a labeled source domain to perform well on an unlabeled target domain with some data distribution shift. While many methods have been proposed in the literature, fair and realistic evaluation remains an open question, particularly due to methodological difficulties in selecting hyperparameters in the unsupervised setting. With SKADA-Bench, we propose a framework to evaluate DA methods on diverse modalities, beyond computer vision task that have been largely explored

---

[*]Equal contribution

in the literature. We present a complete and fair evaluation of existing shallow algorithms, including reweighting, mapping, and subspace alignment. Realistic hyperparameter selection is performed with nested cross-validation and various unsupervised model selection scores, on both simulated datasets with controlled shifts and real-world datasets across diverse modalities, such as images, text, biomedical, and tabular data. Our benchmark highlights the importance of realistic validation and provides practical guidance for real-life applications, with key insights into the choice and impact of model selection approaches. `SKADA-Bench` is open-source, reproducible, and can be easily extended with novel DA methods, datasets, and model selection criteria without requiring re-evaluating competitors. The code is available at https://github.com/scikit-adaptation/skada-bench.

## 1 Introduction

Given some training –or *source*– data, supervised learning consists in estimating a function that makes good predictions on *target* data. However, performance often drops when the source distribution used for training differs from the target distribution used for testing. This shift can be due, for instance, to the collection process or non-stationarity in the data, and is ubiquitous in real-life settings. It has been observed in various application fields, including tabular data (Gardner et al., 2023), clinical data (Harutyunyan et al., 2019), or computer vision (Ganin et al., 2016b).

**Domain adaptation.** Unsupervised Domain Adaptation (DA) addresses this problem by adapting a model trained on a labeled source dataset –or *domain*– so that it performs well on an unlabeled target domain, assuming some distribution shifts between the two (Ben-David et al., 2006; Quinonero-Candela et al., 2008; Redko et al., 2022). As illustrated in Figure 1, source and target distributions can exhibit various types of shifts (Moreno-Torres et al., 2012): changes in feature distributions (covariate shift), class proportions (target shift), conditional distributions (conditional shift), or in distributions in particular subspaces (subspace shift). Depending on the type of shift, existing DA methods attempt to align the source distribution closer to the target using reweighting (Sugiyama & Müller, 2005; Shimodaira, 2000), mapping (Sun et al., 2017; Courty et al., 2017b), or dimension reduction (Pan et al., 2011; Fernando et al., 2013) methods. More recently, it has been proposed to mitigate shifts in a feature space learned by deep learning (Ganin et al., 2016b; Sun & Saenko, 2016; Long et al., 2015a; Damodaran et al., 2018b), primarily focusing on computer vision applications. Regardless of the core algorithm used to address the domain shift, hyperparameters must be tuned for optimal performance. Indeed, a critical challenge in applying DA methods to real-world cases is selecting the appropriate method and tuning its hyperparameters, especially given the unknown shift type and the absence of labels in the target domain.

**Model selection in DA settings.** Without distribution shifts, classical model selection strategies – including hyperparameter optimization– rely on evaluating the generalization error with an independent labeled validation set. However, in DA, validating the hyperparameters in a supervised manner on the target domains is impossible due to the lack of labels. While it is possible to validate the hyperparameters on the source domain, it generally leads to a suboptimal model selection because of the distribution shift. In the literature, this problem is often raised but not always addressed. Some papers choose not to validate the parameters (Pan et al., 2011), while others validate on the source domain (Sun et al., 2017) or propose custom cross-validation methods (Sugiyama et al., 2007b). Few papers focus specifically on DA model selection criteria, which we will call *scorers* in this paper. These scorers are used to select the methods' hyperparameters, and mainly consists of reweighting methods on source (Sugiyama et al., 2007a; You et al., 2019), prediction entropy (Morerio et al., 2017; Saito et al., 2021) or circular validation (Bruzzone & Marconcini, 2010a). One of the goals of our benchmark is to evaluate these approaches in diverse and realistic scenarios.

**Benchmarks of DA.** As machine learning continues to flourish, new methods constantly emerge, making it essential to develop benchmarks that facilitate fair comparisons (Hutson, 2018; Pineau et al., 2019; Mattson et al., 2020; Moreau et al., 2022). In DA and related fields, several benchmarks have been proposed. Numerous papers focus on Out-of-distribution (OOD) datasets for different modalities: computer vision, text, graphs (Koh et al., 2021; Sagawa et al., 2022), time-series (Gagnon-Audet et al., 2023), AI-aided drug discovery (Ji et al., 2023) or tabular dataset (Gardner et al., 2023). Due to the type of data considered, existing

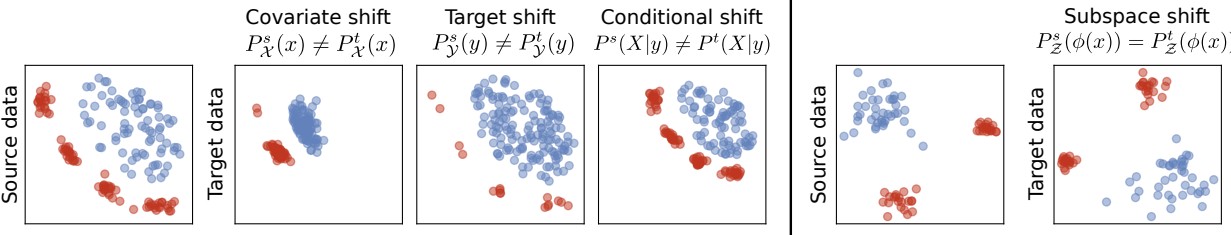

Figure 1: Illustration of different type of distribution shift between source and target domains: covariate shift, target shift, conditional shift, and subspace shift (Mathematial details are available in Section 2.1). Points represent data samples, with colors indicating different classes. These synthetic datasets are used to evaluate model performance under controlled shift scenarios in the experiement part.

benchmarks are mainly focused on Deep DA methods (Musgrave et al., 2021; Wang, 2018; Jiang et al., 2022; Fawaz et al., 2023), offering an incomplete evaluation of DA literature. Moreover, only a few benchmarks propose a comparison of Deep unsupervised DA methods with realistic parameters selection, on computer vision (Hu et al., 2023; Musgrave et al., 2021) and time series (Fawaz et al., 2023) data. Those benchmarks have shown the importance of validating with unsupervised scores and reveal that Deep DA methods achieve much lower performance in realistic scenarios.

**Contributions.** In the following, we propose `SKADA-Bench`, an ambitious and fully reproducible benchmark with the following features: **1.** A set of 4 simulated and 8 real-life datasets with different modalities (computer vision, NLP, tabular, biomedical) totaling 51 realistic shift scenarios, **2.** A wide range of 20 Shallow DA methods designed to handle different types of shifts, **3.** An evaluation of 7 deep DA methods on 4 real-world datasets from the computer vision and biomedical modalities, **4.** A realistic model selection procedure using 5 different unsupervised scorers with nested cross-validation for hyperparameter selection, **5.** An open-source implementation and publicly available datasets, easy to extend for new DA methods and datasets without the need to re-run the whole experiment.

In addition, we provide a detailed analysis of the results and derive guidelines for practitioners to select the best methods depending on the type of shifts, and the best scorer to perform unsupervised model selection. In particular, the effects of model selection and the scorer's choice on the final performances are highlighted, showing a clear gap between the unsupervised realistic scorers versus using target labels for supervised validation.

## 2 Domain adaptation and model selection without target labels

In this section, we first discuss the specificities of the unsupervised domain adaptation problem and introduce several types of data shifts and their corresponding DA methods. Next, we discuss the different validation strategies used in the literature and the need for realistic scorers to compare DA methods.

### 2.1 Data shifts and DA strategies

**Domain Adaptation problem and theory.** The theoretical framework of DA is well established (Ben-David et al., 2006; Quinonero-Candela et al., 2008; Redko et al., 2022). The main results highlight that the performance discrepancy of an estimator between the source and target domains is linked to the divergence between both distributions. This has motivated the majority of DA methods to search for a universal (or domain invariant) predictor by minimizing the divergence between the two domains through the adaptation of the distributions. This is done in practice by modeling and estimating the shift between the source and target distributions and then compensating for this shift before training a predictor.

**Data Shifts and DA methods.** A wide variety of shifts between the source and target distributions are possible. They are usually expressed as a relation between the joint distributions $P^s(x, y) = P^s(x|y)P^s_{\mathcal{Y}}(y) = P^s(y|x)P^s_{\mathcal{X}}(x)$ in the source domain and $P^t(x, y) = P^t(x|y)P^t_{\mathcal{Y}}(y) = P^t(y|x)P^t_{\mathcal{X}}(x)$ in the target domain. We now discuss the main types of shifts and the strategies proposed in the literature to mitigate them. Figure 1

illustrates these shifts.

In **Covariate shift** the conditionals probabilities are equal (*i.e.*, $P^s(y|x) = P^t(y|x)$), but the feature marginals change (*i.e.*, $P^s_{\mathcal{X}}(x) \neq P^t_{\mathcal{X}}(x)$). **Target shift** is similar, but the label marginals change $P^s_{\mathcal{Y}}(y) \neq P^t_{\mathcal{Y}}(y)$ while the conditionals are preserved. For classification problems, it corresponds to a change in the proportion of the classes between the two domains. Both of those shifts can be compensated by **reweighting methods** that assign different weights to the samples of the source domain to make it closer to the target domain (Sugiyama & Müller, 2005; Shimodaira, 2000).

In **Conditional shift**, conditional probabilities differ between domain (*i.e.*, $P^s(x|y) \neq P^t(x|y)$ or $P^s(y|x) \neq P^t(y|x)$). This shift is typically harder to compensate for, necessitating explicit modeling to address it effectively. For instance, several approaches model the shift as a **mapping** $m$ between the source and target domain such that $P^s(y|m(x)) = P^t(y|x)$ (Sun et al., 2017; Courty et al., 2017b). The estimated mapping is then applied to the source data before training a predictor.

**Subspace shift** assumes that while probabilities are different between the domains ($P^s_{\mathcal{X}}(x) \neq P^t_{\mathcal{X}}(x)$ and $P^s(x|y) \neq P^t(x|y)$), there exists a subspace $\mathcal{Z}$ and a function $\phi : \mathcal{X} \to \mathcal{Z}$ such that $P^s_{\mathcal{Z}}(\phi(x)) = P^t_{\mathcal{Z}}(\phi(x))$ and $P^s(y|\phi(x)) = P^t(y|\phi(x))$. Note that this means the shift occurs in the orthogonal complement of $\mathcal{Z}$. This implies that a classifier trained on $\mathcal{Z}$ will perform well across both domains. **Subspace methods** are specifically designed towards identifying the subspace $\mathcal{Z}$ and the function $\phi$, as developed in Pan et al. (2011); Fernando et al. (2013). Note that, as discussed in the introduction, a natural extension of this idea is to learn an invariant feature space using Deep learning (Ganin et al., 2016b; Sun & Saenko, 2016).

## 2.2 DA model selection strategies

As seen above, DA methods are typically designed to correct a specific type of shift. However, in real-world scenarios, the nature of the shift is often unknown. This presents a challenge in selecting the appropriate method and tuning its parameters when facing a new problem. In this section, we discuss the validation strategies proposed in the literature to compare DA methods, focusing on realistic scorers that do not use target labels.

**Realistic DA scorers.** In the literature, few papers propose realistic DA scorers to validate the parameters of the methods, *i.e.*, unsupervised scorers that **do not require target labels**. The *Importance Weighted (IW)* scorer (Sugiyama et al., 2007a) computes the score as a reweighted accuracy on labeled sources data. The *Deep Embedded Validation (DEV)* (You et al., 2019) can be seen as an IW in the latent space with a variance reduction strategy. DEV was originally proposed for Deep learning models but can be used on shallow DA methods that compute features from the data (mapping/subspaces). The *Prediction Entropy (PE)* scorer (Morerio et al., 2017) measures the uncertainty associated with model predictions on the target data. *Soft Neighborhood Density (SND)* (Saito et al., 2021) also computes an entropy but on a normalized pairwise similarity matrix between probabilistic predictions on target. The *Circular Validation (CircV)* scorer (Bruzzone & Marconcini, 2010a) performs DA by first adapting the model from the source to the target domain and predicting the target labels. Next, it adapts back from the target to the source using these estimated labels. Performance is measured as the accuracy between the recovered and true source labels.

The *MixVal* scorer (Hu et al., 2023) also performs domain adaptation by first adapting the model from the source to the target domain and predicting the target labels. Then, it generates mixed target samples by probing intra-cluster samples to assess neighborhood density and inter-cluster samples to examine classification boundaries. The score is the accuracy between the generated targets labels and their predictions to evaluate the consistency.

**DA validation in the literature.** The model selection problem in DA has been widely discussed in the literature. Yet, this literature constitutes a subfield of DA and has seldom been used to validate new DA methods. Indeed, there is no consensus on the best validation strategy and many papers do not properly validate their methods, leading to over-estimated performances. Some authors do not discuss the validation procedure (Sugiyama & Müller, 2005; Shimodaira, 2000) or consider fixed hyperparameters (Huang et al., 2006). While some methods rely on custom validation techniques (Sugiyama et al., 2007b), others use cross-validation, either on the source or the target (Sun et al., 2017; Courty et al., 2017b), or alternatively other validation strategies proposed in the literature (Courty et al., 2017a; Bruzzone & Marconcini, 2010a). A complete picture of the model selection procedures used to validate the methods considered in SKADA-Bench

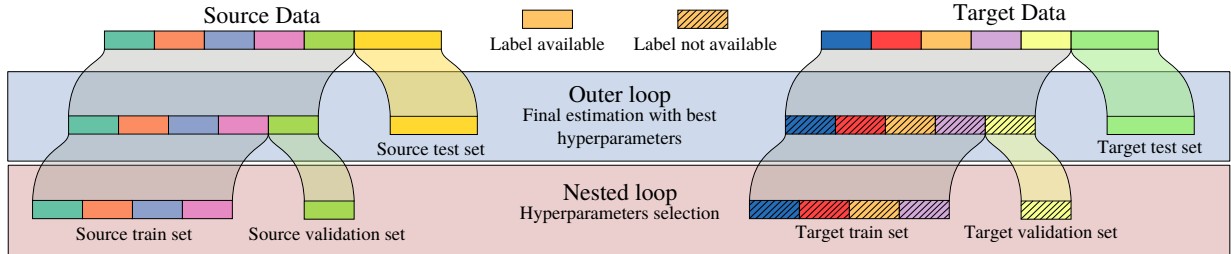

Figure 2: Visualization of nested cross-validation strategy. Both source and target data are split into an outer loop and then a nested loop. The nested loop tunes hyperparameters for the domain adaptation method, while the outer loop trains a final classifier with the best hyperparameters and evaluates its accuracy on both source and target data. Note: Target sets have no labels during the nested loop, reflecting unsupervised Domain Adaptation.

in their original papers is presented in Table 4 in Appendix B. The goal of `SKADA-Bench` is therefore to constitute a dedicated benchmark to compare scorers from the literature and report performances that can be expected in real use cases for the considered methods.

## 3 A realistic benchmark for DA

In this section, we present our benchmark framework. First, we introduce the parameter validation strategies. Then, we present the compared DA methods followed by a description of the datasets used in the benchmark.

### 3.1 Nested cross-validation loop and implementation

We discuss below the nested cross-validation and the implementation details of the benchmark.

**Hyperparameter validation loop.** We propose a nested loop cross-validation procedure, depicted in Figure 2. First, the source and target data are split into multiple outer test and train sets (outer loop in Figure 2). The test sets are kept to compute the final accuracy for both the source and target domains. For each split in the outer loop, we use a nested loop to select the DA methods' parameters. Here, the training sets are further divided into nested train and validation sets (nested loop in Figure 2). Note that no labels are available for the target nested train and validation sets in this loop. The target training set is used to train the DA method, while the target validation set allows to compute the unsupervised score and select the best model.
For both loops, the data is split randomly 5 times using stratified sampling with an 80%/20% train/test split, except for Deep DA methods, where only one split is computed for the outer loop due to computation time. For one given method, we evaluate all the unsupervised scorers discussed earlier, as well as a supervised scorer that uses target labels, over all the nested splits. After averaging, the scores over the splits, the best hyperparameters are selected according to each scorer and then used to train a final classifier on the outer training sets. Although the supervised scorer cannot be used in practice, it is included in our results to actually evaluate the performance drop due to the absence of target labels. To limit complexity and perform a fair comparison of the methods, we set a timeout of 4 hours for performing the nested loop. Additionally, we chose not to use the `CircV` scorer for Deep DA methods, as training neural networks twice is computationally expensive.

**Base estimators and neural networks.** Existing shallow domain adaptation methods typically rely on either a base estimator trained on the adapted data or an iterative estimation process to adapt this estimator to the target data. The choice of the base estimator is crucial, as it significantly impacts the final performance. Before validating the hyperparameters of the DA methods, we determined the best estimator for each dataset using a grid-search on the source data. We tested multiple hyperparameters for Logistic Regression, SVM with RBF kernel, and XGBoost (Chen & Guestrin, 2016), selecting the ones that maximize the average

accuracy on the source test sets. Note that for some methods that specifically require an SVM estimator (*i.e.,* JDOT and DASVM), we only validate SVM as the base estimator. We validated the base estimator separately from the DA methods parameters to reduce computational complexity and avoid too complex hyperparameter grids that can compromise the reliability of DA scorers. For Deep DA methods, we similarly select an appropriate architecture and experimental setup for training on the source data for each dataset: a two-layer convolutional neural network for `MNIST/USPS`, a ResNet50 (He et al., 2016) pretrained on ImageNet (Deng et al., 2009) for `Office31` and `Office Home`, and a ShallowFBCSPNet (Schirrmeister et al., 2017) for `BCI`. These architectures are widely used and well-supported in the literature of computer vision (Musgrave et al., 2021) and BCI (Schirrmeister et al., 2017). During the nested loop, only the DA parameters for each method are validated.

**Best scorer selection and statistical test.** For all methods, we select the best validation scorer as the one that maximizes the averaged accuracy on the target domains for all real datasets. This provides a reasonable and actionable choice of scorer for each DA method for practitioners. For all methods and datasets, we perform a paired Wilcoxon signed-rank test at the 0.05 level to detect significant gain or drop in performance with respect to the no DA approach, denoted by "Train Src" in the following. The test is done using the accuracy measures of the DA method with the selected scorer and the Train Src for all shifts and outer splits, ensuring between 10 and 60 values depending on the dataset. Note that these statistical tests are not performed on Deep DA methods, as the number of splits is too limited for meaningful testing.

**Python implementation.** The benchmark code is available on GitHub upon publication of the paper.[*] Our benchmark is implemented following the `benchopt` framework (Moreau et al., 2022), which provides standardized ways of organizing and running benchmarks for ML in Python. This framework facilitates reproducing the benchmark's results, with tools to install the dependencies, run the methods in parallel, or cache the results to prevent redundant computations. It also makes it easy to extend the benchmark with additional datasets and methods, enabling it to evolve to account for the advances in the field. In the supplementary materials, we provide examples demonstrating how to add DA methods or datasets to the benchmark. Using this framework, we aim to make `SKADA-Bench` a reference benchmark to evaluate new DA methods in realistic scenarios with valid performance estimations.

## 3.2 Compared DA methods

In this section, we present the different families of domain adaptation methods that we compare in our benchmark. The shallow methods are grouped into four categories: reweighting methods, mapping methods, subspace methods, and others. For Deep DA methods, we consider three domain invariant feature methods. We provide a brief description of each method and the corresponding references.

**Reweighting methods.** These methods aim to reweight the source data to make it closer to the target data. The weights are estimated using different methods such as kernel density estimation (Dens. RW) (Sugiyama & Müller, 2005), Gaussian estimation (Gauss. RW) (Shimodaira, 2000), discriminative estimation (Discr. RW) (Shimodaira, 2000), or nearest-neighbors (NN RW) (Loog, 2012). Other reweighting estimate weights by minimizing a divergence between the source and target distributions such as Kullback-Leibler Importance Estimation Procedure (KLIEP) (Sugiyama et al., 2007b) or Kernel Mean Matching (KMM) (Huang et al., 2006). Finally, we also include the MMDTarS method (Zhang et al., 2013) that uses a Maximum Mean Discrepancy (MMD) to estimate the weights under the target shift hypothesis.

**Mapping methods.** These methods aim to find a mapping between the source and target data that minimizes the distribution shift. The Correlation Alignment method (CORAL) (Sun et al., 2017) aligns the second-order statistics of source and target distributions. The Maximum Mean Discrepancy (MMD-LS) method (Zhang et al., 2013) minimizes the MMD to estimate an affine Location-Scale mapping. Finally, the Optimal Transport (OT) mapping methods (Courty et al., 2017b) use the optimal transport plan to align with a non-linear mapping of the source and target distributions with exact OT (MapOT), entropic regularization (EntOT), or class-based regularization (ClassRegOT). Finally, the Linear OT method (Flamary et al., 2020) uses a linear mapping to align the source and target distributions, assuming Gaussian distributions.

---

[*]Our code is available in supplementary materials.

**Subspace methods.** These methods aim to learn a subspace where the source and target data have the same distribution. The Transfer Component Analysis (TCA) method (Pan et al., 2011) searches for a kernel embedding that minimizes the MMD divergence between the domains while preserving the variance. The Subspace Alignment (SA) method (Fernando et al., 2013) aims to learn a subspace where the source and target have their covariance matrices aligned. The Transfer Subspace Learning (TSL) method (Si et al., 2010) aims to learn a subspace using classical supervised loss functions on the source (*e.g.,* PCA, Fisher LDA) but regularized so that the source and target data have the same distribution once projected on the subspace. Finally, the Joint Principal Component Analysis (JPCA) method is a simple baseline that concatenates source and target data before applying a PCA.

**Others.** We also include other methods that do not fit into the previous categories. The Domain Adaptation SVM (DASVM) method (Bruzzone & Marconcini, 2010a) is a self-labeling method that iteratively updates SVM estimators by adding new target samples with predicted labels and removing source samples. The Joint Distribution Optimal Transport (JDOT) method (Courty et al., 2017a) aims to learn a target predictor that minimizes an OT loss between the joint source and target distributions. The Optimal Transport Label Propagation (OTLabelProp) method (Solomon et al., 2014) uses the optimal transport plan to propagate labels from the source to the target domain.

**Deep DA methods.** These methods aim to reduce the divergence between the source and target data distributions within the learned feature space while simultaneously learning a classifier on source data. The training loss consists in a traditional supervised loss on labeled source data and a second term measuring the discrepancy between the source and target distributions. The methods implemented in the Deep DA Benchmark use different discrepancies, such as covariance distance (Sun & Saenko, 2016) for DeepCORAL, adversarial loss (Ganin et al., 2016a; Zhang et al., 2019) for DANN or MDD, MMD distance (Long et al., 2015a) for DAN, optimal transport distance (Damodaran et al., 2018a) for DeepJDOT, class confusion (Jin et al., 2020) for MCC, and graph spectral alignment (Xiao et al., 2023b) for SPA. Note that these approaches are not part of the main shallow DA benchmark but have been added to provide an interesting comparison of DA performances between shallow and Deep methods on computer vision and biomedical data.

## 3.3 Compared datasets

In this section, we present the datasets used in our experiments. We first introduce the synthetic datasets that implement different known shifts. Then, we describe the real-world datasets from various modalities and tasks such as Computer Vision (CV), Natural language Processing (NLP), tabular data, and biosignals.

**Simulated datasets.** The objective of the simulated datasets is to evaluate the performance of the DA methods under different types of shifts. Knowing that multiple DA methods have been built to handle specific shifts, evaluating them with this dataset will demonstrate whether they perform as expected and if they are properly validated.
The four simulated shifts in 2D, covariate (Cov. shift), target (Tar. shift) conditional (Cond. shift) and Subspace (Sub. shift) shift are illustrated in Figure 1. The source domain is represented by two non-linearly separable classes generated from one large and several smaller Gaussian blobs. In the experiments, the level of noise has been adjusted from Figure 1 to make the problem more difficult. For the subspace shift scenario, the source domain consists of one class represented by a large Gaussian blob and another class comprising Gaussian blobs positioned along the sides of the large one. The target domain is flipped along the diagonal, making the task challenging in the original space but feasible upon diagonal projection.

**Real-word datasets.** The real-world datasets used in our benchmark are summarized in Table 1. We select 8 datasets from different modalities and tasks: Computer Vision (CV) with `Office31` (Koniusz et al., 2017), `Office Home` (Venkateswara et al., 2017), and `MNIST/USPS` (Liao & Carneiro, 2015), Natural Language Processing (NLP) with `20Newsgroup` (Lang, 1995) and `Amazon Review` (McAuley et al., 2015), Tabular Data with `Mushrooms` (Dai et al., 2007) and `Phishing` (Mohammad et al., 2012), and Biosignals with `BCI Competition IV` (Tangermann et al., 2012). The datasets are chosen to represent a wide range of shifts and to evaluate the performance of the methods on different types of data.
Data are preprocessed to extract relevant features, while keeping computational costs reasonable. Images are embedded using deep pre-trained models (except `MNIST/USPS` where the images are vectorized), and textual

Table 1: Characteristics of the real-world datasets used in `SKADA-Bench`.

| Dataset | Modality | Preprocessing | # adapt | # classes | # samples | # features |
|---|---|---|---|---|---|---|
| Office 31 (Koniusz et al., 2017) | CV | Decaff + PCA (Donahue et al., 2014) | 6 | 31 | 470 ± 350 | 100 |
| Office Home (Venkateswara et al., 2017) | CV | ResNet + PCA (He et al., 2016) | 12 | 65 | 3897 ± 850 | 100 |
| MNIST/USPS (Liao & Carneiro, 2015) | CV | Vect + PCA | 2 | 10 | 3000 / 10000 | 50 |
| 20 Newsgroup (Lang, 1995) | NLP | LLM + PCA (Reimers & Gurevych (2019), Xiao et al. (2023a)) | 6 | 2 | 3728 ± 174 | 50 |
| Amazon Review (McAuley & Leskovec (2013), McAuley et al. (2015)) | NLP | LLM + PCA (Reimers & Gurevych (2019), Xiao et al. (2023a)) | 12 | 4 | 2000 | 50 |
| Mushrooms (Dai et al., 2007) | Tabular | One Hot Encoding | 2 | 2 | 4062 ± 546 | 117 |
| Phishing (Mohammad et al., 2012) | Tabular | NA | 2 | 2 | 5527 ± 1734 | 30 |
| BCI (Tangermann et al., 2012) | Biosignals | Cov+TS (Barachant et al., 2012) | 9 | 4 | 288 | 253 |

data is embedded using Large Language Models (LLM) (Reimers & Gurevych, 2019; Xiao et al., 2023a). The tabular data are one-hot encoded to transform categorical data into numerical data. The biosignals from Brain-Computer Interface (`BCI`) data are embedded using the state-of-the-art tangent space representation proposed in Barachant et al. (2012). For images and text, we apply a PCA to reduce the dimensionality of the embeddings to a reasonable dimension to avoid big computational costs. For Deep DA methods, only 4 datasets are used: `Office31`, `Office Home`, `MNIST/USPS` and `BCI`. Since these methods focus on learning feature representations, the data are used in their raw form. The datasets are split into pairs of source and target domains totaling 51 adaptation tasks in the benchmark. More details about the datasets and pre-processing are available in Appendix C.

## 4 Benchmark results

We now present the results of the benchmark. Training and evaluation across all shallow experiments required 1,215 CPU-hours on a standard Slurm (Yoo et al., 2003) cluster, while the Deep DA experiments required 244 GPU-hours. We first discuss and compare the performances of the methods on the different datasets. Then, a detailed study of the unsupervised scorers is provided.

### 4.1 Performance of the DA methods

**Results table.** First, we report the realistic performances of the different methods when using their selected scorer on the different datasets in Table 2. The cells showcasing a significant change in performance with the Wilcoxon test are highlighted with colors. Blue indicates an increase in performance, while red indicates a loss. The intensity of the color corresponds to the magnitude of the gain or loss - the darker the shade, the larger the positive or negative change. Cells with a NA values indicate that the method was not applicable to the dataset (DASVM is limited to binary classification) or that the method has reached a timeout. We also report the best scorer and the average rank of the methods for all real datasets. In addition to Table 2 providing realistic performance estimations with the best realistic scorer, we also report in Table 20 (Appendix E) the results when using the non-realistic supervised scorer.

**Simulated data with known shifts.** DA methods tend to show a significant gain on the shift they were designed for. It is especially true for mapping methods which greatly outperforms the Train Src approach under conditional shift (Cond. shift), almost reaching the Train Tgt performance for EntOT and ClassRegOT. The results also highlight that the mapping methods struggle with target shift (Tar. shift), which is a well-known limitation of this kind of approach (Redko et al., 2019). On the contrary, reweighting methods provide robust performance on target shift. Regarding covariatie shift (Cov. shift), the improvement with reweighting methods is very limited although reweighting is specifically designed for this kind of shift. We believe that using a complex base estimator (here an SVM with an RBF kernel) enables us to train an estimator that works well on both source and target, reducing the impact of importance weighting as previously highlighted

Table 2: Accuracy score for all datasets compared for all the shallow methods for simulated and real-life datasets. The color indicates the amount of the improvement. A white color means the method is not statistically different from Train Src (Train on source). Blue indicates that the score improved with the DA methods, while red indicates a decrease. The darker the color, the more significant the change.

| | | Cov. shift | Tar. shift | Cond. shift | Sub. shift | Office31 | OfficeHome | MNIST/USPS | 20NewsGroups | AmazonReview | Mushrooms | Phishing | BCI | Selected Scorer | Rank |
|---|---|---|---|---|---|---|---|---|---|---|---|---|---|---|---|
| | Train Src | 0.88 | 0.85 | 0.66 | 0.19 | 0.65 | 0.56 | 0.54 | 0.59 | 0.7 | 0.72 | 0.91 | 0.55 | | 10.66 |
| | Train Tgt | 0.92 | 0.93 | 0.82 | 0.98 | 0.89 | 0.8 | 0.96 | 1.0 | 0.73 | 1.0 | 0.97 | 0.64 | | 1.55 |
| Reweighting | Dens. RW | 0.88 | 0.86 | 0.66 | 0.18 | 0.62 | 0.56 | 0.54 | 0.58 | 0.7 | 0.71 | 0.91 | 0.55 | IW | 12.20 |
| | Disc. RW | 0.85 | 0.83 | 0.71 | 0.18 | 0.63 | 0.54 | 0.5 | 0.6 | 0.68 | 0.75 | 0.91 | 0.56 | CircV | 8.75 |
| | Gauss. RW | 0.89 | 0.86 | 0.65 | 0.21 | 0.22 | 0.44 | 0.11 | 0.54 | 0.55 | 0.51 | 0.46 | 0.25 | CircV | 16.45 |
| | KLIEP | 0.88 | 0.86 | 0.66 | 0.19 | 0.65 | 0.56 | 0.54 | 0.6 | 0.69 | 0.72 | 0.91 | 0.55 | CircV | 10.56 |
| | KMM | 0.89 | 0.85 | 0.64 | 0.16 | 0.64 | 0.54 | 0.52 | 0.7 | 0.57 | 0.74 | 0.91 | 0.52 | CircV | 11.74 |
| | NN RW | 0.89 | 0.86 | 0.67 | 0.15 | 0.65 | 0.55 | 0.54 | 0.59 | 0.66 | 0.71 | 0.91 | 0.54 | CircV | 9.15 |
| | MMDTarS | 0.88 | 0.86 | 0.64 | 0.2 | 0.6 | 0.56 | 0.54 | 0.59 | 0.7 | 0.74 | 0.91 | 0.55 | IW | 10.81 |
| Mapping | CORAL | 0.74 | 0.7 | 0.76 | 0.18 | 0.65 | 0.57 | 0.62 | 0.73 | 0.7 | 0.72 | 0.92 | 0.62 | CircV | 5.08 |
| | MapOT | 0.72 | 0.57 | 0.82 | 0.02 | 0.6 | 0.51 | 0.61 | 0.76 | 0.68 | 0.63 | 0.84 | 0.47 | PE | 10.21 |
| | EntOT | 0.71 | 0.6 | 0.82 | 0.12 | 0.64 | 0.58 | 0.6 | 0.83 | 0.62 | 0.75 | 0.86 | 0.54 | CircV | 9.40 |
| | ClassRegOT | 0.74 | 0.58 | 0.81 | 0.11 | NA | 0.53 | 0.62 | 0.97 | 0.68 | 0.82 | 0.89 | 0.52 | IW | 8.25 |
| | LinOT | 0.73 | 0.73 | 0.76 | 0.18 | 0.66 | 0.57 | 0.64 | 0.82 | 0.7 | 0.76 | 0.91 | 0.61 | CircV | 4.06 |
| | MMD-LS | 0.78 | 0.72 | 0.76 | 0.56 | 0.65 | 0.56 | 0.55 | 0.97 | 0.63 | 0.85 | NA | 0.5 | MixVal | 8.22 |
| Subspace | JPCA | 0.88 | 0.85 | 0.66 | 0.15 | 0.62 | 0.48 | 0.51 | 0.77 | 0.69 | 0.78 | 0.9 | 0.54 | PE | 8.98 |
| | SA | 0.74 | 0.68 | 0.8 | 0.11 | 0.65 | 0.57 | 0.56 | 0.88 | 0.67 | 0.78 | 0.89 | 0.53 | CircV | 7.80 |
| | TCA | 0.52 | 0.47 | 0.51 | 0.62 | 0.04 | 0.02 | 0.07 | 0.61 | 0.61 | 0.49 | 0.48 | 0.26 | DEV | 17.58 |
| | TSL | 0.88 | 0.85 | 0.66 | 0.2 | 0.63 | 0.48 | 0.45 | 0.63 | 0.69 | 0.45 | 0.89 | 0.26 | IW | 15.09 |
| Other | JDOT | 0.72 | 0.58 | 0.82 | 0.13 | 0.6 | 0.42 | 0.59 | 0.79 | 0.67 | 0.65 | 0.79 | 0.47 | IW | 11.42 |
| | OTLabelProp | 0.72 | 0.59 | 0.8 | 0.07 | 0.66 | 0.56 | 0.62 | 0.86 | 0.67 | 0.64 | 0.86 | 0.5 | CircV | 10.01 |
| | DASVM | 0.89 | 0.86 | 0.65 | 0.15 | NA | NA | NA | 0.87 | NA | 0.83 | 0.85 | NA | MixVal | 7.29 |

in (Byrd & Lipton, 2019) for deep neural networks. The results reported in Table 17 of Appendix E reveal that reweighting methods significantly outperform Train Src when using a linear base classifier.

**Real data with unknown shift.** *The performance of reweighting methods is often close to Train Src baseline on real datasets.* For example, NN RW reaches only a 54% accuracy on MNIST/USPS compared to 54% for Train Src. This result can be be due to the violations of the same-support assumption, which is crucial for reweighting to work effectively (Segovia-Martín et al., 2023), and which is likely true for the three CV datasets. In this case, hyperparameter tuning frequently select configurations leading to near-uniform weighting, which explain the close performance to Train Src.
*The performance of mapping methods is dataset-dependent, potentially due to the number of classes and presence of target shift.* Mapping methods excel on MNIST/USPS and 20NewsGroup which respectively contain 10 and 2 classes, but fail on Office31 and OfficeHome with 31 and 60 classes. For example, ClassRegOT achieves 62% on MNIST/USPS and 97% on 20NewsGroup, but only 53% on OfficeHome. Additionally, while mapping performs well on the NLP dataset 20NewsGroup, it results in negative transfer on Amazon Reviews which has target shifts.
*It is notable that simple transformations are the best in average across all modalities.* Indeed, most methods that significantly outperform Train Src in ranking average across all modalities are LinOT, CORAL, JPCA, and SA, which all rely on linear transformations such as scaling, linear projection, or rotations. These methods are robust across datasets and modalities, offering effective alignment with minimal risk of negative DA. For instance, LinOT consistently ranks among the top 5 methods and achieves substantial gains over Train Src: +10 points on MNIST/USPS (64% vs. 54%), +23 points on 20NewsGroup (82% vs. 59%), and +6 points on BCI (61% vs. 55%).

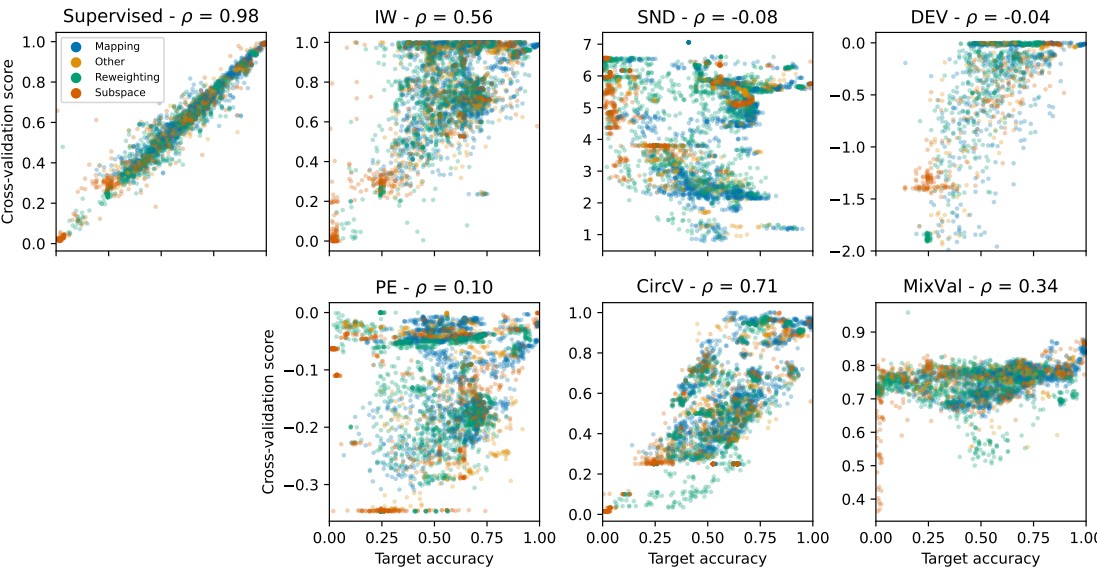

Figure 3: Cross-val score as a function of the accuracy for different supervised and unsupervised scorers. The Pearson correlation coefficient is reported for each scorer by $\rho$. Each point represents an inner split with a DA method (color of the points) and a dataset. A good score should correlate with the target accuracy.

**Computational cost.** While full computational results are reported in Appendix E.7 and Figure 11, we briefly highlight that several top-performing methods—such as LinOT, CORAL, and SA—also exhibit some of the lowest training and testing times, averaging under 20 seconds per task. This reinforces their appeal in practice: they combine competitive performance with high efficiency, making them especially suitable for settings with limited computational resources or many adaptation tasks.

**Take-away for DA users.** Reweighting methods are best suited for scenarios where the same-support assumption holds and perform particularly well when paired with regularized hypotheses like linear models. Even when assumptions are not fully satisfied, reweighting tends to be robust to negative transfer. Mapping methods are highly effective under moderate numbers of classes and in the absence of target shift but carry a significant risk of negative transfer if target shift is present. When the type of distribution shift is uncertain, simpler transformation-based methods like LinOT, CoRAL, JPCA, and SA provide modest performance improvements while minimizing risks of negative DA, making them reliable and safe default options.

**Selected scorer per DA method.** We observe that the best scorer differs across methods, Circular Validation has been selected 10 out of 20 times as the best scorer, followed by Importance Weighting 4 out of 20 times. Table 20 in the supplementary material provides the non-realistic accuracy results with the supervised scorer. It is worth noting that the supervised scorer generally outperforms the unsupervised ones. For example, EntOT achieves +5 points with the supervised scorer versus with the CircV scorer (88% vs 83%) on the 20NewsGroups dataset. It is crucial to choose the model realistically to avoid producing overly optimistic results, as many data analysis papers have done (see Table 4).

These results show the methods' sensitivity to parameter selections and the difficulty of using realistic scorers. This might also explain why DA methods are not widely used in practice: they are very difficult to tune and might decrease performances compared with no adaptation.

## 4.2 Study of validation scorers

We now investigate the performance of the various scorers to select hyperparameters of the DA method. First, we consider the relationship between the cross-val score and the accuracy for each inner split. In Figure 3, we plot for each scorer the cross-val score as a function of the accuracy computed on the test set and report the Pearson correlation coefficient $\rho$. As expected, the supervised scorer is highly correlated with the accuracy ($\rho = 0.98$), as it has access to the target labels. We observe that SND, DEV, and PE do not provide a

good proxy to select hyperparameters that give the best-performing models ($\rho \leq 0.06$). On the contrary, MixVal, IW and CircV are correlated with the accuracy, $\rho = 0.34$, $\rho = 0.56$ and $\rho = 0.71$ respectively. This is coherent with their selection as the best scorer in most scenarios in Table 2. Still, while those scorers are well correlated with the target accuracy, it is important to note that they have a large variance. For instance, a score close to 1 in IW or CircV corresponds to an accuracy between 0.5 and 1.0.

Furthermore, we provide in Figures 9 and 10, from Appendix E, several visualizations that illustrate the relationship between the accuracy achieved when using a supervised scorer and the accuracy obtained when using different unsupervised scorers. We also visualize in Figure 8 the drop in performance when using the best-unsupervised scorer instead of the supervised scorer. Interestingly some methods such as KMM, EntOT, and ClassRegOT can lose up to 10% accuracy when using realistic scorers, which might come from their higher number of parameters or their sensitivity to them.

Our results thus show that most scorers have poor results when evaluated on many datasets. Of the six methods under consideration, only two achieve satisfactory performance, although incurring large variance in their results. This shows that proper hyperparameter selection is still an open question, that needs attention from the research comunity to guide practitioners toward real life applications of unsupervised DA technics.

### 4.3 Deep DA methods

Although most of the recent work on domain adaptation focus on Deep methods for computer vision tasks, shallow methods are competitive in many applications such as tabular data (Grinsztajn et al., 2022) or datasets with a relatively small number of training examples such as BCI (Chevallier et al., 2024). Moreover, shallow methods can also benefit from recent advances in Deep learning by using Deep pre-trained feature extraction (transfer learning). However, to the best of our knowledge,

Table 3: Accuracy scores for Deep methods on selected real-life datasets using DA scorers. LinOT is reported as the overall top-performing shallow method. Green indicates that the score improved with the DA methods. The darker the color, the more significant the change.

| | MNIST/USPS | Office31 | OfficeHome | BCI | Selected Scorer | Rank |
|---|---|---|---|---|---|---|
| Train Src | 0.85 | 0.77 | 0.58 | 0.54 | | 6.19 |
| Train Tgt | 0.98 | 0.96 | 0.83 | 0.56 | | 2.07 |
| DeepCORAL (Sun & Saenko, 2016) | 0.93 | 0.77 | 0.59 | 0.54 | MixVal | 3.29 |
| DAN (Long et al., 2015b) | 0.86 | 0.75 | 0.56 | 0.53 | IW | 4.76 |
| DANN (Ganin et al., 2016a) | 0.9 | 0.79 | 0.59 | 0.41 | MixVal | 4.98 |
| DeepJDOT (Damodaran et al., 2018b) | 0.9 | 0.82 | 0.62 | 0.54 | PE | 2.92 |
| MCC (Jin et al., 2020) | 0.93 | 0.83 | 0.66 | 0.53 | MixVal | 2.38 |
| MDD (Zhang et al., 2019) | 0.87 | 0.78 | 0.56 | 0.4 | MixVal | 4.96 |
| SPA (Xiao et al., 2023b) | 0.91 | 0.78 | 0.56 | 0.41 | DEV | 5.39 |
| LinOT (Flamary et al., 2020) | 0.64 | 0.6 | 0.57 | 0.61 | CircV | |

the literature lacks quantitative comparison between shallow methods applied on Deep pre-trained feature extraction and Deep DA methods. To this end, we ran a benchmark using the same pipeline as in Table 2 with three Domain Invariant Deep DA methods on the CV and BCI datasets.

**Results table.** The results are available in Table 3 with a comparison to the best performing shallow method from Table 2. One of the most notable and expected difference is on MNIST/USPS. Shallow methods struggle to achieve good performances, even on Train Tgt, as they rely on PCA for feature extraction. Deep methods, on the other hand, use CNNs, leading to large accuracy gains on train on Src and Tgt but also on Deep DA methods. However, it is important to note that while DeepJDOT, DANN and MCC improve performance on all datasets, they remain far from the train on Tgt accuracies, partly due to the difficulty in tuning their parameters (see Appendix E.3 with the supervised scorer). The superior performance of Deep DA methods on CV datasets can be attributed to the relationship between classification in the DA subspace and the disentanglement of semantic (discriminant) content from style (domain shift) (Gonzalez-Garcia et al., 2018; Gabbay et al., 2021). Numerous studies have demonstrated that semantic embeddings can be effectively recovered, supporting the assumption that a (nonlinear) subspace shift is reasonable for CV tasks. However, for the BCI dataset, where the amount of data is limited, the performances of Deep DA methods are inferior to some other shallow methods (*i.e.,* LinOT for example). Finally, a method like DANN, which is often considered as a baseline in the community, has been shown to be hard to validate and requires setup that can be difficult to determine across different settings. These results emphasize, that while Deep invariant DA

methods can be effective, they do not consistently yield good results across modalities, whereas shallow DA methods can achieve similar or superior performances with less effort and fewer computational resources in low data regimes.

**Limitations and future work.** The evaluation of deep DA methods in this benchmark is limited to a single outer split to ensure computational feasibility. While practical, this setting may introduce variance in performance estimates, especially on small datasets. Moreover, although all deep methods completed within the allocated time, the 4-hour timeout may restrict the exploration of more complex architectures or longer training procedures on larger datasets.

We view these constraints as a limitation of the current benchmark and an opportunity for future work. A comprehensive evaluation of deep DA methods would benefit from larger datasets ($> 10^4$ samples) with more stable training dynamics, and lower-variance evaluation using a single train/validation/test split. Extending DA-Bench to support such large-scale evaluations represents a promising direction for improving the practical assessment of deep DA approaches.

## 5 Conclusion

In this work, we introduced `SKADA-Bench`, a extensive benchmark for unsupervised domain adaptation, carefully evaluating the impact of the model selection criteria and covering diverse modalities: computer vision, natural language processing, tabular data and biosignals. While being quite comprehenvise on shallow methods, our results also provide a comparison of three common deep DA baselines on computer vision and biosignals. Importantly `SKADA-Bench` can be easily extended with new datasets and methods to push further the state-of-the-art. Our findings reveal that few shallow DA methods consistently perform well across diverse datasets and that model selection scorers significantly influence their effectiveness. While deep DA methods show similar trends, they often require more extensive hyperparameter tuning and architectures tailored to each modality. Notably, they tend to perform significantly better than shallow methods on some modalities, such as computer vision, while facing challenges on others such as biosignals. For each DA method, we provide the optimal model selection scorer for unsupervised hyperparameter tuning based on our experiments.

## Acknowledgments

This work benefited from state aid managed by the Agence Nationale de la Recherche under the France 2030 programme, reference ANR-23-IACL-0005, ANR-22-PESN-0012 and ANR-20-CHIA-0016. This research was also supported in part by the French National Research Agency (ANR) through the MATTER project (ANR-23-ERCC-0006-01) and the BenchArk project (ANR-24-IAS2-0003). It received funding from the Fondation de l'École polytechnique. Additionally, this project received funding from the European Union's Horizon Europe research and innovation program under grant agreement 101120237 (ELIAS).

All the datasets used for this work were accessed and processed on the Inria and IP Paris (IDCS) compute infrastructures.

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

## A   Appendix

# Appendix

**Reproducibility.** The entire code and results of `SKADA-Bench` are open-sourced at https://github.com/scikit-adaptation/skada-bench. The implementation of the DA methods and scorers is provided along with access to the simulated and real-world datasets. All the performance tables and figures can be reproduced effortlessly, and guidelines with minimal working examples are given to add new DA methods and datasets.

**Roadmap.** In this appendix, we provide additional information regarding the validation procedure used in the literature for each DA method implemented in `SKADA-Bench` in Section B. We provide a detailed description of the data and preprocessing used in `SKADA-Bench` in Section C. In Section D, we give minimal working Python examples to add a new DA method and dataset in `SKADA-Bench`. Finally, we provide the detailed benchmark results in Section E. In particular, the results per dataset can be found in Section E.1. We discuss in Section E.2 the impact of the choice of base estimator on the performance of DA methods for the simulated datasets. The results of each DA method with the supervised scorer on all the datasets are given in Table 20 of Section E.3, which parallels Table 2. A thorough analysis of the effect of using realistic unsupervised scorers is also provided in Section E.6. Finally, the computational efficiency of each DA method is studied in Section E.7 and the hyperparameters used for grid search are given in Section E.8. We display the corresponding table of contents below.

## Table of Contents

# B  Model selection in Domain Adaptation

Table 4: Validation procedure in Domain Adaptation methods. NA stands for *not applicable* and means that there are no hyperparameters. None means that no validation procedure has been conducted or that it is not specified in the original paper.

| | Method | Validation Procedure | Comment |
|---|---|---|---|
| Reweighting | Density Reweight (Sugiyama & Müller, 2005) | None | Bandwidth fixed by Silverman method |
| | Discriminative Reweight (Shimodaira, 2000) | NA | No hyperparameters |
| | Gaussian Reweight (Shimodaira, 2000) | None | Not specified in (Shimodaira, 2000) |
| | KLIEP (Sugiyama et al., 2007b) | Integrated CV | Likelihood CV (Sugiyama et al., 2007b) on target |
| | KMM (Huang et al., 2006) | None | Fixed data-dependent hyperparameters |
| | NN Reweight (Loog, 2012) | None | Number of neighbors fixed to one |
| | MMDTarS (Zhang et al., 2013) | CV | Not specified if done on source or target |
| Mapping | Coral (Sun et al., 2017) | NA | No hyperparameters |
| | OT mapping (Courty et al., 2017b) | CV target/CircCV | Unclear in the text |
| | Lin. OT mapping (Flamary et al., 2020) | NA | No hyperparameters |
| | MMD-LS (Zhang et al., 2013) | CV | Not specified if done on source or target |
| Subsp. | SA (Fernando et al., 2013) | 2-fold CV on source | - |
| | TCA (Pan et al., 2011) | Validation on target | Target subset used to tune parameters |
| | TSL (Si et al., 2010) | None | Not specified in (Si et al., 2010) |
| Other | JDOT (Courty et al., 2017a) | Reverse CV (Zhong et al., 2010) | - |
| | OT label prop (Solomon et al., 2014) | NA | No hyperparameters |
| | DASVM (Bruzzone & Marconcini, 2010a) | Circular Validation (Bruzzone & Marconcini, 2010a) | - |

In Table 4, we provide additional information on the validation procedures used in the original papers that proposed the different domain adaptation methods implemented in `SKADA-Bench`. The first column is the name of the method, the second column contains the procedure used to select hyperparameters and the last column provides additional details. What is striking is that many methods do not conduct or specify a validation procedure to select the hyperparameters, which limits the performance of the proposed method on a novel dataset. Several others rely on cross-validation using target data. However, since target labels are typically unavailable in practical scenarios, this validation approach is unrealistic. Overall, many methods have been evaluated with unrealistic or not reproducible validation procedures, making the performance of the proposed methods appear over-optimistic. A key contribution of our work is the extensive comparison of realistic, unsupervised scorers for selecting optimal hyperparameters and base estimators in DA methods.

## C Datasets description and preprocessing

The simulated dataset proves that DA methods can work well under the proper shift (see Table 2). However, in real-world applications, we do not have prior knowledge of the type of data shift. Hence, finding the appropriate domain-adaptation method between reweighting, mapping, and subspace methods is a challenging task. In this section, we introduce 8 real-world datasets coming from different fields. Table 1 summarizes the 8 classification datasets used in this benchmark with the corresponding data modality, preprocessing, number of source-target pairs (# adapt), number of classes, samples, and feature dimensions.

**Computer Vision.** First, three computer vision datasets are proposed: Office31 (Koniusz et al., 2017), Office Home (Venkateswara et al., 2017), and MNIST/USPS (Liao & Carneiro, 2015). We create embeddings for Office31 using the Decaff preprocessing method (Donahue et al., 2014) and for Office Home using a pre-trained ResNet50 (He et al., 2016). These embeddings, as well as vectorized MNIST/USPS, are dimensionally reduced with a Principal Component Analysis (PCA). These three datasets encompass 3, 4, and 3 domains, respectively and all pairs of adaptations are used as DA problems. MNIST/USPS contain clear and blurry images digits, Office31 differentiates between images captured by various devices, while for OfficeHome, its by image style.

**NLP.** The second task is Natural Language Processing (NLP). Two datasets are studied: 20Newsgroup (Lang, 1995) and Amazon Review (McAuley et al., 2015). The 20Newsgroup dataset contains 20.000 documents categorized into 4 categories: *talk*, *rec*, *comp*, and *sci*. The learning task is to classify documents across categories. First, the documents are embedded using a Large Language Model (LLM) (Reimers & Gurevych, 2019; Xiao et al., 2023a), and then PCA is applied for dimensionality reduction.

For the Amazon Review dataset, the task is to classify comment ratings. This dataset spans four domains (Books, DVDs, Kitchen, Electronics), and the domain shift results from these varying types of objects. Similar to the 20Newsgroup dataset, comments are embedded using the same LLM and then reduced in dimensionality using a PCA.

**Tabular data.** We propose two tabular datasets. The first one is the Mushroom dataset (Dai et al., 2007), where the task is to classify whether a mushroom is poisonous or not. The two domains are separated according to the mushroom's stalk shape (enlarging vs. tapering). The tabular data are one-hot-encoded to transform categorical data into numerical data. The second dataset is Phishing (Mohammad et al., 2012). The classification problem involves determining whether a webpage is a phishing or a legitimate one. The domains are separated according to the availability of the IP address. Since the data are already numerical, no preprocessing is done on this dataset.

**Biosignals.** The last task is BCI Motor Imagery. The dataset used is BCI Competition IV (Tangermann et al., 2012), often used in the literature (Barachant et al., 2012) and availaoble in MOABB (Aristimunha et al., 2023). The task is to classify four kinds of motor imagery (right hand, left hand, feet, and tongue) from EEG data. In this dataset, nine subjects are available. The domains are separated based on session number. For each subject, session 1 is considered as the source domain and session 2 is considered as the target domain. The data are multivariate signals. To embed the data, we first compute the covariance and then project this covariance on the Tangent Space as proposed in Barachant et al. (2012).

## D Adding new methods and datasets to `SKADA-Bench`

Using the `benchopt` framework for this benchmark allows users to easily add novel domain adaptation (DA) methods and datasets. To that end, users should adhere to the `benchopt` (Moreau et al., 2022) conventions. We provide below the guidelines with examples in Python to add a new DA method and a new dataset to `SKADA-Bench`.

### D.1 Adding a new DA method

A new DA method can be easily added with the following:

- Create file with a class called `Solver` that inherits from `DASolver` and place it in the `solvers` folder.

- This class should implement a `get_estimator()` function, which returns a class inheriting from `sklearn.BaseEstimator` and accepts `sample_weight` as fit parameter. In the benchmark we used the Domain Adaptation toolbox SKADA (Gnassounou et al., 2024) that provides many DA estimatos with correct interface.

We provide below an example of Python implementation to add a new DA method to `SKADA-Bench`.

```python
# Python snippet code to add a DA method
from benchmark_utils.base_solver import DASolver
from sklearn.base import BaseEstimator

class MyDAEstimator(BaseEstimator):
    def __init__(self, param1=10, param2='auto'):
        self.param1 = param1
        self.param2 = param2

    def fit(self, X, y, sample_weight=None):
        # sample_weight<0 are source samples
        # sample_weight>=0 are target samples
        # y contains -1 for masked target samples
        # Your code here : store stuff in self for later predict
        return self

    def predict(self, X):
        # do prediction on target domain here
        return ypred

    def predict_proba(self, X):
        # do probabilistic prediction on target domain here
        return proba

class Solver(DASolver):
    name = "My_DA_method"

    # Param grid to validate
    default_param_grid = {
        'param1': [10, 100],
        'param2': ['auto', 'manual']
    }

    def get_estimator(self):
        return MyDAEstimator()
```

### D.2 Adding a new dataset

A new DA dataset can be easily added with the following:

- Create a file with a class called `Dataset` that inherits from `BaseDataset` and place it in the `datasets` folder.

- This class should implement a `get_data()` function, which returns a dictionary with keys `X`, `y`, and `sample_domain`.

We provide below an example of Python implementation to add a new dataset to `SKADA-Bench`.

```python
# Python snippet code to add a dataset

from benchopt import BaseDataset
from sklearn.datasets import make_blobs
import numpy as np

class Dataset(BaseDataset):
    name = "example_dataset"

    def get_data(self):
        X_source, y_source = make_blobs(
        n_samples=100, centers=3,
        n_features=2, random_state=0
        )

        X_target, y_target = make_blobs(
        n_samples=100, centers=5,
        n_features=2, random_state=42
        )
        # sample_domain>0 for source samples
        # sample_domain<0 for target samples
        sample_domain = np.array([1]*len(X_source) + [-2]*len(X_target))

        return dict(
            X=np.concatenate((X_source, X_target), axis=0)
            y=np.concatenate((y_source, y_target))
            sample_domain=sample_domain
        )
```

By following these guidelines, users can seamlessly integrate their own datasets and DA methods into `SKADA-Bench`. It results in a user-friendly benchmark that enables fast, reproducible, and reliable comparisons of common and novel DA methods and datasets. We will provide users with precomputed result files and utilities, allowing them to run only the new methods or datasets. This will speed up new comparisons and avoid unnecessary computations.

## E   Benchmark detailed results

### E.1   Results per datasets

In Table 2 of the main paper, the reported performance for each method on a given dataset is an average over the number of shifts, i.e., the number of source-target pairs denoted by #adapt in Table 1. In this section, we provide additional details on the performance of methods for each shift in each dataset. These results are presented in separate tables for each dataset

These detailed tables where cell in green denote a gain wrt Train Src (average outside of standard deviation of Train Src) better illustrate the challenges of domain adaptation (DA) methods. They show that not all shifts are equivalent within a given dataset. For example, Table 12 reveals that only 3 shifts in the AmazonReview dataset present a DA problem (defined as a $> 3\%$ difference in accuracy between Train Src and Train Tgt). While for the other shifts, we achieve similar performance whether we train on source

or target data. Additionally, some specific shifts present a DA problem that no method can successfully address. This can be seen in the dsl → amz shift in the Office31 dataset, as shown in Table 7. Finally, some DA methods perform consistently across all shifts within a dataset, as demonstrated by the results for the 20Newsgroup dataset in Table 11.

Table 5: Accuracy score for MNIST/USPS dataset for each shift compared for all the methods. A white color means the method does not increase the performance compared to Train Src (Train on the source). Green indicates that the performance improved with the DA methods. The darker the color, the more significant the change.

| | | MNIST→USPS | USPS→MNIST | Mean | Rank |
|---|---|---|---|---|---|
| | Train Src | 0.66 ± 0.02 | 0.43 ± 0.02 | 0.54 ± 0.02 | 12.00 |
| | Train Tgt | 0.96 ± 0.0 | 0.96 ± 0.01 | 0.96 ± 0.01 | 1.00 |
| Reweighting | Dens. RW | 0.66 ± 0.02 | 0.42 ± 0.02 | 0.54 ± 0.02 | 13.25 |
| | Disc. RW | 0.6 ± 0.02 | 0.4 ± 0.02 | 0.5 ± 0.02 | 19.00 |
| | Gauss. RW | 0.11 ± 0.01 | 0.11 ± 0.01 | 0.11 ± 0.01 | 20.00 |
| | KLIEP | 0.66 ± 0.02 | 0.43 ± 0.02 | 0.54 ± 0.02 | 13.25 |
| | KMM | 0.64 ± 0.02 | 0.41 ± 0.03 | 0.52 ± 0.02 | 18.00 |
| | NN RW | 0.66 ± 0.02 | 0.42 ± 0.02 | 0.54 ± 0.02 | 12.00 |
| | MMDTarS | 0.66 ± 0.02 | 0.42 ± 0.02 | 0.54 ± 0.02 | 12.75 |
| Mapping | CORAL | 0.74 ± 0.01 | 0.51 ± 0.01 | 0.62 ± 0.01 | 5.50 |
| | MapOT | 0.69 ± 0.02 | 0.54 ± 0.02 | 0.61 ± 0.02 | 4.00 |
| | EntOT | 0.66 ± 0.02 | 0.54 ± 0.02 | 0.6 ± 0.02 | 5.00 |
| | ClassRegOT | 0.66 ± 0.01 | 0.53 ± 0.06 | 0.59 ± 0.04 | 11.50 |
| | LinOT | 0.74 ± 0.02 | 0.53 ± 0.02 | 0.64 ± 0.02 | 3.25 |
| | MMD-LS | 0.66 ± 0.02 | 0.47 ± 0.02 | 0.56 ± 0.02 | 8.25 |
| Subspace | JPCA | 0.66 ± 0.02 | 0.43 ± 0.02 | 0.54 ± 0.02 | 12.00 |
| | SA | 0.71 ± 0.03 | 0.36 ± 0.11 | 0.54 ± 0.07 | 12.00 |
| | TCA | 0.08 ± 0.07 | 0.11 ± 0.02 | 0.09 ± 0.05 | 21.00 |
| | TSL | 0.66 ± 0.02 | 0.43 ± 0.02 | 0.54 ± 0.02 | 10.50 |
| Other | JDOT | 0.73 ± 0.02 | 0.53 ± 0.02 | 0.63 ± 0.02 | 3.50 |
| | OTLabelProp | 0.71 ± 0.03 | 0.53 ± 0.02 | 0.62 ± 0.02 | 6.50 |

Table 6: Accuracy score for MNIST/USPS dataset for each shift compared for all the Deep DA methods. A white color means the method does not increase the performance compared to Train Src (Train on the source). Green indicates that the performance improved with the DA methods. The darker the color, the more significant the change.

| | MNIST→USPS | USPS→MNIST | Mean | Rank |
|---|---|---|---|---|
| Train Src | 0.94 | 0.76 | 0.85 | 5.0 |
| Train Tgt | 0.99 | 0.99 | 0.99 | 1.0 |
| DANN | 0.94 | 0.88 | 0.91 | 4.0 |
| DeepCORAL | 0.97 | 0.89 | 0.93 | 2.5 |
| DeepJDOT | 0.96 | 0.9 | 0.93 | 2.5 |

Table 7: Accuracy score for Office31 dataset for each shift compared for all the methods. A white color means the method does not increase the performance compared to Train Src (Train on the source). Green indicates that the performance improved with the DA methods. The darker the color, the more significant the change.

| | | amz→dsl | amz→web | dsl→amz | dsl→web | web→amz | Mean | Rank |
|---|---|---|---|---|---|---|---|---|
| | Train Src | 0.55 ± 0.04 | 0.51 ± 0.03 | 0.5 ± 0.02 | 0.9 ± 0.02 | 0.49 ± 0.01 | 0.59 ± 0.02 | 8.40 |
| | Train Tgt | 0.94 ± 0.03 | 0.95 ± 0.01 | 0.78 ± 0.02 | 0.95 ± 0.02 | 0.77 ± 0.01 | 0.88 ± 0.02 | 1.00 |
| Reweighting | Dens. RW | 0.49 ± 0.05 | 0.51 ± 0.04 | 0.5 ± 0.02 | 0.85 ± 0.09 | 0.49 ± 0.01 | 0.57 ± 0.04 | 11.40 |
| | Disc. RW | 0.56 ± 0.06 | 0.49 ± 0.03 | 0.49 ± 0.02 | 0.89 ± 0.03 | 0.46 ± 0.01 | 0.58 ± 0.03 | 13.30 |
| | Gauss. RW | 0.34 ± 0.03 | 0.11 ± 0.04 | 0.05 ± 0.02 | 0.38 ± 0.04 | 0.11 ± 0.03 | 0.2 ± 0.03 | 19.80 |
| | KLIEP | 0.55 ± 0.05 | 0.5 ± 0.04 | 0.49 ± 0.02 | 0.91 ± 0.02 | 0.48 ± 0.01 | 0.59 ± 0.03 | 10.60 |
| | KMM | 0.56 ± 0.06 | 0.54 ± 0.04 | 0.49 ± 0.03 | 0.87 ± 0.03 | 0.47 ± 0.02 | 0.59 ± 0.03 | 10.20 |
| | NN RW | 0.54 ± 0.04 | 0.5 ± 0.03 | 0.49 ± 0.02 | 0.91 ± 0.01 | 0.48 ± 0.01 | 0.58 ± 0.02 | 12.30 |
| | MMDTarS | 0.53 ± 0.04 | 0.52 ± 0.04 | 0.5 ± 0.03 | 0.81 ± 0.14 | 0.49 ± 0.01 | 0.57 ± 0.05 | 10.00 |
| Mapping | CORAL | 0.53 ± 0.04 | 0.53 ± 0.03 | 0.5 ± 0.03 | 0.91 ± 0.03 | 0.5 ± 0.01 | 0.6 ± 0.03 | 6.20 |
| | MapOT | 0.47 ± 0.03 | 0.5 ± 0.04 | 0.47 ± 0.01 | 0.85 ± 0.03 | 0.43 ± 0.01 | 0.54 ± 0.03 | 13.30 |
| | EntOT | 0.52 ± 0.03 | 0.56 ± 0.02 | 0.5 ± 0.03 | 0.88 ± 0.02 | 0.46 ± 0.01 | 0.58 ± 0.02 | 11.80 |
| | ClassRegOT | 0.58 ± 0.07 | 0.59 ± 0.04 | 0.51 ± 0.02 | 0.84 ± 0.04 | 0.25 ± 0.25 | 0.55 ± 0.08 | 9.20 |
| | LinOT | 0.54 ± 0.05 | 0.56 ± 0.05 | 0.49 ± 0.02 | 0.9 ± 0.02 | 0.49 ± 0.01 | 0.6 ± 0.03 | 8.90 |
| | MMD-LS | 0.55 ± 0.04 | 0.51 ± 0.04 | 0.49 ± 0.03 | 0.9 ± 0.02 | 0.48 ± 0.01 | 0.59 ± 0.03 | 8.00 |
| Subspace | JPCA | 0.53 ± 0.04 | 0.51 ± 0.02 | 0.49 ± 0.02 | 0.89 ± 0.01 | 0.48 ± 0.01 | 0.58 ± 0.02 | 14.10 |
| | SA | 0.42 ± 0.17 | 0.5 ± 0.03 | 0.5 ± 0.02 | 0.84 ± 0.09 | 0.49 ± 0.02 | 0.55 ± 0.07 | 12.10 |
| | TCA | 0.03 ± 0.01 | 0.03 ± 0.0 | 0.04 ± 0.0 | 0.04 ± 0.0 | 0.04 ± 0.0 | 0.03 ± 0.0 | 20.80 |
| | TSL | 0.55 ± 0.04 | 0.51 ± 0.04 | 0.49 ± 0.03 | 0.9 ± 0.02 | 0.49 ± 0.01 | 0.59 ± 0.03 | 9.20 |
| Other | JDOT | 0.64 ± 0.05 | 0.63 ± 0.02 | 0.5 ± 0.02 | 0.77 ± 0.04 | 0.49 ± 0.01 | 0.61 ± 0.03 | 6.20 |
| | OTLabelProp | 0.58 ± 0.03 | 0.63 ± 0.02 | 0.49 ± 0.04 | 0.89 ± 0.02 | NA | 0.65 ± 0.03 | 8.75 |

Table 8: Accuracy score for Office31 dataset for each shift compared for all the deep DA methods. A white color means the method does not increase the performance compared to Train Src (Train on the source). Green indicates that the performance improved with the DA methods. The darker the color, the more significant the change.

| | amz→dsl | amz→web | dsl→amz | dsl→web | web→amz | web→dsl | Mean | Rank |
|---|---|---|---|---|---|---|---|---|
| Train Src | 0.72 | 0.75 | 0.61 | 0.94 | 0.63 | 0.99 | 0.77 | 4.17 |
| Train Tgt | 0.99 | 1.0 | 0.87 | 0.99 | 0.88 | 0.99 | 0.95 | 1.25 |
| DANN | 0.75 | 0.8 | 0.64 | 0.96 | 0.62 | 0.98 | 0.79 | 3.75 |
| DeepCORAL | 0.77 | 0.77 | 0.61 | 0.98 | 0.63 | 0.99 | 0.79 | 3.33 |
| DeepJDOT | 0.79 | 0.8 | 0.68 | 0.97 | 0.69 | 1.0 | 0.82 | 2.17 |

Table 9: Accuracy score for OfficeHome dataset for each shift compared for all the methods. A white color means the method does not increase the performance compared to Train Src (Train on the source). Green indicates that the performance improved with the DA methods. The darker the color, the more significant the change.

| Group | Method | art→clipart | art→product | art→realworld | clipart→art | clipart→product | clipart→realworld | product→art | product→clipart | product→realworld | realworld→art | realworld→clipart | realworld→product | Mean | Rank |
|---|---|---|---|---|---|---|---|---|---|---|---|---|---|---|---|
| | Train Src | 0.34 ± 0.02 | 0.63 ± 0.01 | 0.7 ± 0.01 | 0.46 ± 0.02 | 0.6 ± 0.01 | 0.63 ± 0.01 | 0.5 ± 0.02 | 0.36 ± 0.02 | 0.71 ± 0.01 | 0.63 ± 0.01 | 0.38 ± 0.01 | 0.74 ± 0.02 | 0.56 ± 0.01 | 10.42 |
| | Train Tgt | 0.71 ± 0.02 | 0.92 ± 0.01 | 0.84 ± 0.01 | 0.73 ± 0.03 | 0.91 ± 0.01 | 0.83 ± 0.01 | 0.72 ± 0.01 | 0.7 ± 0.01 | 0.83 ± 0.01 | 0.75 ± 0.02 | 0.7 ± 0.01 | 0.91 ± 0.01 | 0.8 ± 0.01 | 1.00 |
| Reweighting | Dens. RW | 0.34 ± 0.02 | 0.63 ± 0.01 | 0.7 ± 0.01 | 0.47 ± 0.02 | 0.58 ± 0.06 | 0.63 ± 0.01 | 0.5 ± 0.02 | 0.36 ± 0.02 | 0.71 ± 0.01 | 0.64 ± 0.01 | 0.37 ± 0.03 | 0.73 ± 0.05 | 0.55 ± 0.02 | 9.71 |
| Reweighting | Disc. RW | 0.32 ± 0.01 | 0.56 ± 0.03 | 0.71 ± 0.02 | 0.42 ± 0.02 | 0.52 ± 0.01 | 0.57 ± 0.02 | 0.53 ± 0.01 | 0.35 ± 0.02 | 0.72 ± 0.01 | 0.63 ± 0.01 | 0.37 ± 0.01 | 0.73 ± 0.01 | 0.54 ± 0.02 | 13.00 |
| Reweighting | Gauss. RW | 0.22 ± 0.01 | 0.43 ± 0.01 | 0.59 ± 0.02 | 0.35 ± 0.01 | 0.46 ± 0.03 | 0.48 ± 0.02 | 0.41 ± 0.02 | 0.25 ± 0.02 | 0.61 ± 0.02 | 0.53 ± 0.02 | 0.26 ± 0.02 | 0.67 ± 0.02 | 0.44 ± 0.02 | 18.08 |
| Reweighting | KLIEP | 0.34 ± 0.02 | 0.63 ± 0.01 | 0.7 ± 0.01 | 0.46 ± 0.02 | 0.6 ± 0.01 | 0.63 ± 0.01 | 0.5 ± 0.02 | 0.36 ± 0.02 | 0.71 ± 0.01 | 0.64 ± 0.01 | 0.38 ± 0.01 | 0.74 ± 0.03 | 0.56 ± 0.02 | 9.00 |
| Reweighting | KMM | 0.32 ± 0.02 | 0.62 ± 0.01 | 0.7 ± 0.01 | 0.45 ± 0.07 | 0.61 ± 0.02 | 0.64 ± 0.02 | 0.48 ± 0.01 | 0.34 ± 0.02 | 0.7 ± 0.01 | 0.58 ± 0.01 | 0.35 ± 0.04 | 0.73 ± 0.01 | 0.54 ± 0.02 | 13.96 |
| Reweighting | NN RW | 0.34 ± 0.02 | 0.63 ± 0.01 | 0.7 ± 0.02 | 0.45 ± 0.02 | 0.59 ± 0.01 | 0.61 ± 0.01 | 0.5 ± 0.02 | 0.35 ± 0.02 | 0.71 ± 0.01 | 0.63 ± 0.01 | 0.37 ± 0.01 | 0.74 ± 0.02 | 0.55 ± 0.02 | 12.96 |
| Reweighting | MMDTarS | 0.34 ± 0.02 | 0.63 ± 0.01 | 0.7 ± 0.01 | 0.46 ± 0.02 | 0.61 ± 0.01 | 0.63 ± 0.01 | 0.5 ± 0.02 | 0.36 ± 0.02 | 0.71 ± 0.01 | 0.64 ± 0.01 | 0.38 ± 0.01 | 0.74 ± 0.03 | 0.56 ± 0.02 | 7.71 |
| Mapping | CORAL | 0.39 ± 0.02 | 0.64 ± 0.01 | 0.7 ± 0.0 | 0.48 ± 0.02 | 0.58 ± 0.0 | 0.61 ± 0.01 | 0.54 ± 0.02 | 0.38 ± 0.01 | 0.72 ± 0.01 | 0.64 ± 0.01 | 0.43 ± 0.01 | 0.75 ± 0.02 | 0.57 ± 0.01 | 6.62 |
| Mapping | MapOT | 0.36 ± 0.03 | 0.53 ± 0.01 | 0.64 ± 0.01 | 0.41 ± 0.02 | 0.52 ± 0.01 | 0.53 ± 0.01 | 0.45 ± 0.01 | 0.37 ± 0.02 | 0.64 ± 0.01 | 0.59 ± 0.02 | 0.41 ± 0.01 | 0.67 ± 0.02 | 0.51 ± 0.02 | 12.42 |
| Mapping | EntOT | 0.38 ± 0.02 | 0.65 ± 0.01 | 0.72 ± 0.01 | 0.5 ± 0.01 | 0.61 ± 0.02 | 0.66 ± 0.02 | 0.55 ± 0.02 | 0.38 ± 0.01 | 0.71 ± 0.02 | 0.65 ± 0.01 | 0.42 ± 0.01 | 0.73 ± 0.01 | 0.58 ± 0.01 | 5.00 |
| Mapping | ClassRegOT | 0.39 ± 0.02 | 0.66 ± 0.01 | 0.73 ± 0.01 | 0.53 ± 0.01 | 0.63 ± 0.01 | 0.67 ± 0.01 | 0.56 ± 0.03 | 0.39 ± 0.01 | 0.72 ± 0.02 | 0.64 ± 0.01 | 0.43 ± 0.01 | 0.74 ± 0.02 | 0.59 ± 0.01 | 3.42 |
| Mapping | LinOT | 0.39 ± 0.02 | 0.64 ± 0.02 | 0.7 ± 0.01 | 0.49 ± 0.02 | 0.59 ± 0.0 | 0.62 ± 0.01 | 0.54 ± 0.02 | 0.38 ± 0.01 | 0.72 ± 0.01 | 0.63 ± 0.01 | 0.43 ± 0.01 | 0.75 ± 0.03 | 0.57 ± 0.01 | 6.54 |
| Mapping | MMD-LS | 0.34 ± 0.02 | 0.63 ± 0.02 | 0.7 ± 0.01 | 0.46 ± 0.02 | 0.6 ± 0.01 | 0.62 ± 0.01 | 0.5 ± 0.02 | 0.36 ± 0.02 | 0.71 ± 0.01 | 0.63 ± 0.02 | 0.38 ± 0.01 | 0.74 ± 0.02 | 0.56 ± 0.02 | 6.46 |
| Subspace | JPCA | 0.34 ± 0.02 | 0.63 ± 0.02 | 0.7 ± 0.01 | 0.46 ± 0.02 | 0.6 ± 0.01 | 0.63 ± 0.01 | 0.5 ± 0.02 | 0.35 ± 0.02 | 0.71 ± 0.01 | 0.63 ± 0.01 | 0.38 ± 0.01 | 0.74 ± 0.02 | 0.56 ± 0.01 | 10.58 |
| Subspace | SA | 0.39 ± 0.02 | 0.64 ± 0.01 | 0.7 ± 0.01 | 0.5 ± 0.02 | 0.51 ± 0.21 | 0.63 ± 0.01 | 0.54 ± 0.02 | 0.38 ± 0.02 | 0.71 ± 0.02 | 0.63 ± 0.01 | 0.41 ± 0.0 | 0.75 ± 0.03 | 0.57 ± 0.03 | 6.50 |
| Subspace | TCA | 0.02 ± 0.0 | 0.01 ± 0.0 | 0.02 ± 0.0 | 0.02 ± 0.01 | 0.02 ± 0.0 | 0.02 ± 0.01 | 0.01 ± 0.01 | 0.02 ± 0.01 | 0.02 ± 0.01 | 0.03 ± 0.01 | 0.02 ± 0.0 | 0.02 ± 0.01 | 0.02 ± 0.01 | 19.17 |
| OtLabelProp | JDOT | NA | NA | NA | NA | NA | NA | 0.48 ± 0.01 | NA | 0.65 ± 0.02 | NA | NA | NA | 0.56 ± 0.02 | 16.75 |
| OtLabelProp | OTLabelProp | 0.32 ± 0.02 | 0.57 ± 0.01 | 0.66 ± 0.02 | 0.51 ± 0.02 | 0.63 ± 0.01 | 0.63 ± 0.02 | 0.53 ± 0.01 | 0.39 ± 0.01 | 0.7 ± 0.01 | 0.63 ± 0.02 | 0.43 ± 0.01 | 0.74 ± 0.02 | 0.56 ± 0.01 | 9.25 |

Table 10: Accuracy score for OfficeHome dataset for each shift compared for all the deep DA methods. A white color means the method does not increase the performance compared to Train Src (Train on the source). Green indicates that the performance improved with the DA methods. The darker the color, the more significant the change.

| | art→clipart | art→product | art→realworld | clipart→art | clipart→product | clipart→realworld | product→art | product→clipart | product→realworld | realworld→art | realworld→clipart | realworld→product | Mean | Rank |
|---|---|---|---|---|---|---|---|---|---|---|---|---|---|---|
| Train Src | 0.42 | 0.62 | 0.75 | 0.53 | 0.6 | 0.62 | 0.52 | 0.31 | 0.73 | 0.67 | 0.41 | 0.76 | 0.58 | 4.83 |
| Train Tgt | 0.78 | 0.91 | 0.86 | 0.8 | 0.93 | 0.85 | 0.8 | 0.78 | 0.86 | 0.78 | 0.76 | 0.92 | 0.83 | 1.00 |
| DANN | 0.44 | 0.63 | 0.73 | 0.58 | 0.61 | 0.62 | 0.54 | 0.38 | 0.75 | 0.67 | 0.44 | 0.76 | 0.6 | 4.25 |
| DeepCORAL | 0.47 | 0.64 | 0.75 | 0.63 | 0.59 | 0.65 | 0.59 | 0.39 | 0.76 | 0.7 | 0.45 | 0.78 | 0.62 | 3.00 |
| DeepJDOT | 0.47 | 0.65 | 0.74 | 0.63 | 0.65 | 0.66 | 0.59 | 0.44 | 0.77 | 0.71 | 0.47 | 0.79 | 0.63 | 2.33 |

Table 11: Accuracy score for 20NewsGroups dataset for each shift compared for all the methods. A white color means the method does not increase the performance compared to Train Src (Train on the source). Green indicates that the performance improved with the DA methods. The darker the color, the more significant the change.

| | | rec→sci | rec→talk | sci→rec | sci→talk | talk→rec | talk→sci | Mean | Rank |
|---|---|---|---|---|---|---|---|---|---|
| | Train Src | 0.52 ± 0.03 | 0.56 ± 0.05 | 0.54 ± 0.05 | 0.7 ± 0.01 | 0.52 ± 0.05 | 0.71 ± 0.03 | 0.59 ± 0.04 | 17.58 |
| | Train Tgt | 1.0 ± 0.0 | 1.0 ± 0.0 | 1.0 ± 0.0 | 0.99 ± 0.0 | 1.0 ± 0.0 | 0.99 ± 0.0 | 1.0 ± 0.0 | 1.00 |
| Reweighting | Dens. RW | 0.49 ± 0.03 | 0.56 ± 0.05 | 0.53 ± 0.05 | 0.69 ± 0.03 | 0.52 ± 0.05 | 0.71 ± 0.03 | 0.58 ± 0.04 | 17.67 |
| | Disc. RW | 0.45 ± 0.03 | 0.61 ± 0.07 | 0.44 ± 0.02 | 0.76 ± 0.01 | 0.57 ± 0.04 | 0.78 ± 0.02 | 0.6 ± 0.03 | 15.50 |
| | Gauss. RW | 0.7 ± 0.07 | 0.45 ± 0.0 | 0.66 ± 0.11 | 0.49 ± 0.01 | 0.44 ± 0.0 | 0.48 ± 0.02 | 0.54 ± 0.04 | 17.33 |
| | KLIEP | 0.52 ± 0.03 | 0.56 ± 0.05 | 0.57 ± 0.1 | 0.7 ± 0.01 | 0.52 ± 0.05 | 0.71 ± 0.03 | 0.6 ± 0.05 | 16.50 |
| | KMM | 0.65 ± 0.06 | 0.71 ± 0.03 | 0.7 ± 0.06 | 0.74 ± 0.02 | 0.65 ± 0.14 | 0.75 ± 0.03 | 0.7 ± 0.06 | 12.17 |
| | NN RW | 0.52 ± 0.06 | 0.57 ± 0.06 | 0.54 ± 0.03 | 0.7 ± 0.01 | 0.52 ± 0.06 | 0.71 ± 0.03 | 0.59 ± 0.04 | 16.42 |
| | MMDTarS | 0.52 ± 0.03 | 0.57 ± 0.05 | 0.54 ± 0.05 | 0.7 ± 0.01 | 0.52 ± 0.06 | 0.71 ± 0.03 | 0.59 ± 0.04 | 14.17 |
| Mapping | CORAL | 0.61 ± 0.03 | 0.79 ± 0.02 | 0.62 ± 0.01 | 0.76 ± 0.01 | 0.78 ± 0.01 | 0.78 ± 0.02 | 0.73 ± 0.02 | 11.67 |
| | MapOT | 0.67 ± 0.02 | 0.84 ± 0.01 | 0.68 ± 0.02 | 0.78 ± 0.02 | 0.83 ± 0.01 | 0.79 ± 0.02 | 0.76 ± 0.02 | 7.75 |
| | EntOT | 0.82 ± 0.0 | 0.82 ± 0.02 | 0.78 ± 0.05 | 0.87 ± 0.01 | 0.82 ± 0.02 | 0.87 ± 0.01 | 0.83 ± 0.02 | 6.50 |
| | ClassRegOT | 0.92 ± 0.14 | 0.93 ± 0.1 | 0.98 ± 0.01 | 0.96 ± 0.01 | 0.99 ± 0.0 | 0.97 ± 0.0 | 0.96 ± 0.04 | 2.67 |
| | LinOT | 0.72 ± 0.02 | 0.92 ± 0.0 | 0.72 ± 0.01 | 0.79 ± 0.11 | 0.91 ± 0.01 | 0.83 ± 0.07 | 0.82 ± 0.04 | 6.50 |
| | MMD-LS | 0.99 ± 0.0 | 0.98 ± 0.01 | 0.99 ± 0.0 | 0.95 ± 0.01 | 0.98 ± 0.01 | 0.96 ± 0.01 | 0.98 ± 0.01 | 3.00 |
| Subspace | JPCA | 0.71 ± 0.04 | 0.72 ± 0.06 | 0.73 ± 0.05 | 0.7 ± 0.01 | 0.84 ± 0.02 | 0.71 ± 0.03 | 0.74 ± 0.04 | 11.75 |
| | SA | 0.83 ± 0.03 | 0.84 ± 0.01 | 0.85 ± 0.01 | 0.87 ± 0.01 | 0.84 ± 0.03 | 0.87 ± 0.01 | 0.85 ± 0.02 | 3.83 |
| | TCA | 0.59 ± 0.25 | 0.52 ± 0.08 | NA | NA | 0.55 ± 0.09 | 0.56 ± 0.01 | 0.56 ± 0.11 | 15.50 |
| Other | JDOT | 0.67 ± 0.02 | 0.84 ± 0.01 | 0.68 ± 0.02 | 0.79 ± 0.02 | 0.84 ± 0.01 | 0.8 ± 0.01 | 0.77 ± 0.01 | 8.50 |
| | OTLabelProp | 0.78 ± 0.01 | 0.95 ± 0.01 | 0.78 ± 0.01 | 0.84 ± 0.05 | 0.95 ± 0.01 | 0.83 ± 0.04 | 0.86 ± 0.02 | 5.00 |
| | DASVM | 0.59 ± 0.16 | 0.77 ± 0.02 | 0.56 ± 0.11 | 0.78 ± 0.01 | 0.84 ± 0.05 | 0.79 ± 0.02 | 0.72 ± 0.06 | 10.00 |

Table 12: Accuracy score for AmazonReview dataset for each shift compared for all the methods. A white color means the method does not increase the performance compared to Train Src (Train on the source). Green indicates that the performance improved with the DA methods. The darker the color, the more significant the change.

| | | books→electronics | books→kitchen | dvd→books | dvd→electronics | dvd→kitchen | electronics→books | electronics→dvd | electronics→kitchen | kitchen→books | kitchen→dvd | kitchen→electronics | Mean | Rank |
|---|---|---|---|---|---|---|---|---|---|---|---|---|---|---|
| | Train Src | 0.65 ± 0.03 | 0.71 ± 0.02 | 0.72 ± 0.01 | 0.66 ± 0.02 | 0.72 ± 0.01 | 0.69 ± 0.01 | 0.71 ± 0.01 | 0.75 ± 0.01 | 0.71 ± 0.02 | 0.7 ± 0.02 | 0.71 ± 0.01 | 0.7 ± 0.01 | 3.45 |
| | Train Tgt | 0.71 ± 0.02 | 0.77 ± 0.01 | 0.74 ± 0.01 | 0.69 ± 0.03 | 0.76 ± 0.01 | 0.72 ± 0.01 | 0.72 ± 0.02 | 0.76 ± 0.01 | 0.72 ± 0.01 | 0.72 ± 0.01 | 0.71 ± 0.01 | 0.73 ± 0.01 | 1.00 |
| Reweighting | Dens. RW | 0.65 ± 0.03 | 0.71 ± 0.02 | 0.72 ± 0.01 | 0.65 ± 0.02 | 0.72 ± 0.01 | 0.69 ± 0.01 | 0.71 ± 0.01 | 0.74 ± 0.01 | 0.71 ± 0.02 | 0.7 ± 0.02 | 0.71 ± 0.01 | 0.7 ± 0.01 | 6.82 |
| | Disc. RW | 0.63 ± 0.01 | 0.71 ± 0.01 | 0.72 ± 0.01 | 0.64 ± 0.02 | 0.7 ± 0.01 | 0.68 ± 0.03 | 0.62 ± 0.05 | 0.73 ± 0.01 | 0.7 ± 0.01 | 0.69 ± 0.01 | 0.7 ± 0.01 | 0.68 ± 0.02 | 7.32 |
| | Gauss. RW | 0.53 ± 0.01 | 0.62 ± 0.01 | 0.54 ± 0.06 | 0.38 ± 0.06 | 0.51 ± 0.15 | NA | NA | 0.62 ± 0.01 | NA | NA | NA | 0.53 ± 0.05 | 19.33 |
| | KLIEP | 0.65 ± 0.03 | 0.69 ± 0.04 | 0.72 ± 0.01 | 0.66 ± 0.02 | 0.72 ± 0.01 | 0.69 ± 0.03 | NA | 0.75 ± 0.01 | 0.71 ± 0.01 | 0.68 ± 0.03 | 0.71 ± 0.01 | 0.69 ± 0.02 | 6.30 |
| | KMM | 0.52 ± 0.04 | 0.63 ± 0.01 | 0.6 ± 0.04 | 0.5 ± 0.02 | 0.64 ± 0.04 | 0.28 ± 0.2 | NA | 0.63 ± 0.05 | 0.66 ± 0.03 | 0.64 ± 0.03 | 0.58 ± 0.02 | 0.57 ± 0.05 | 16.10 |
| | NN RW | 0.59 ± 0.05 | 0.69 ± 0.01 | 0.71 ± 0.03 | 0.51 ± 0.1 | 0.66 ± 0.06 | 0.68 ± 0.03 | NA | 0.71 ± 0.04 | 0.7 ± 0.02 | 0.65 ± 0.02 | 0.68 ± 0.02 | 0.66 ± 0.04 | 10.65 |
| | MMDTarS | 0.65 ± 0.03 | 0.7 ± 0.02 | 0.72 ± 0.01 | 0.67 ± 0.01 | 0.72 ± 0.01 | 0.7 ± 0.01 | 0.71 ± 0.02 | 0.74 ± 0.01 | 0.7 ± 0.03 | 0.7 ± 0.02 | 0.71 ± 0.02 | 0.7 ± 0.02 | 7.55 |
| Mapping | CORAL | 0.62 ± 0.03 | 0.72 ± 0.01 | 0.73 ± 0.01 | 0.61 ± 0.01 | 0.72 ± 0.01 | 0.69 ± 0.01 | 0.69 ± 0.01 | 0.73 ± 0.02 | 0.7 ± 0.01 | 0.7 ± 0.01 | 0.7 ± 0.02 | 0.69 ± 0.01 | 4.56 |
| | MapOT | 0.6 ± 0.01 | 0.7 ± 0.0 | 0.71 ± 0.01 | 0.59 ± 0.01 | 0.7 ± 0.0 | 0.68 ± 0.01 | 0.66 ± 0.01 | 0.74 ± 0.01 | 0.69 ± 0.01 | 0.69 ± 0.01 | 0.64 ± 0.01 | 0.67 ± 0.01 | 9.36 |
| | EntOT | 0.53 ± 0.01 | 0.62 ± 0.0 | 0.64 ± 0.0 | 0.53 ± 0.0 | 0.62 ± 0.0 | 0.65 ± 0.0 | 0.66 ± 0.01 | 0.74 ± 0.01 | 0.64 ± 0.0 | 0.64 ± 0.0 | 0.54 ± 0.0 | 0.62 ± 0.0 | 13.86 |
| | ClassRegOT | 0.68 ± 0.02 | 0.72 ± 0.02 | 0.69 ± 0.03 | 0.67 ± 0.02 | 0.72 ± 0.02 | 0.61 ± 0.01 | 0.61 ± 0.02 | 0.69 ± 0.03 | 0.65 ± 0.02 | 0.65 ± 0.02 | 0.7 ± 0.02 | 0.67 ± 0.02 | 9.00 |
| | LinOT | 0.64 ± 0.03 | 0.73 ± 0.02 | 0.73 ± 0.01 | 0.61 ± 0.01 | 0.73 ± 0.0 | 0.69 ± 0.02 | NA | 0.74 ± 0.0 | 0.71 ± 0.02 | 0.7 ± 0.02 | 0.7 ± 0.02 | 0.7 ± 0.02 | 3.60 |
| | MMD-LS | 0.56 ± 0.02 | 0.54 ± 0.03 | 0.72 ± 0.01 | 0.63 ± 0.06 | 0.61 ± 0.16 | 0.65 ± 0.1 | 0.55 ± 0.15 | NA | 0.63 ± 0.11 | 0.48 ± 0.01 | 0.71 ± 0.01 | 0.61 ± 0.07 | 14.30 |
| Subspace | JPCA | 0.61 ± 0.0 | 0.69 ± 0.01 | 0.69 ± 0.01 | 0.63 ± 0.01 | 0.69 ± 0.0 | 0.68 ± 0.01 | 0.68 ± 0.01 | 0.7 ± 0.01 | 0.67 ± 0.01 | 0.68 ± 0.01 | 0.64 ± 0.01 | 0.67 ± 0.01 | 10.14 |
| | SA | 0.62 ± 0.02 | 0.72 ± 0.01 | 0.73 ± 0.02 | 0.62 ± 0.01 | 0.72 ± 0.01 | 0.69 ± 0.01 | 0.69 ± 0.01 | 0.74 ± 0.02 | 0.71 ± 0.02 | 0.69 ± 0.01 | 0.69 ± 0.03 | 0.69 ± 0.02 | 8.27 |
| | TCA | 0.52 ± 0.0 | 0.62 ± 0.0 | 0.64 ± 0.0 | 0.52 ± 0.0 | 0.62 ± 0.0 | 0.64 ± 0.0 | 0.64 ± 0.0 | 0.62 ± 0.0 | 0.64 ± 0.0 | 0.64 ± 0.0 | 0.52 ± 0.0 | 0.6 ± 0.0 | 18.45 |
| | TSL | 0.65 ± 0.03 | 0.71 ± 0.02 | 0.72 ± 0.01 | 0.66 ± 0.02 | 0.72 ± 0.01 | 0.63 ± 0.01 | NA | 0.75 ± 0.01 | 0.71 ± 0.02 | 0.7 ± 0.02 | 0.71 ± 0.01 | 0.7 ± 0.02 | 5.94 |
| Other | JDOT | 0.61 ± 0.01 | 0.71 ± 0.01 | 0.72 ± 0.01 | 0.6 ± 0.01 | 0.7 ± 0.0 | 0.63 ± 0.01 | NA | 0.71 ± 0.02 | 0.69 ± 0.02 | 0.69 ± 0.01 | 0.67 ± 0.03 | 0.67 ± 0.01 | 8.75 |
| | OTLabelProp | 0.6 ± 0.01 | 0.69 ± 0.01 | 0.7 ± 0.01 | 0.59 ± 0.01 | 0.69 ± 0.01 | 0.68 ± 0.02 | 0.69 ± 0.02 | 0.74 ± 0.01 | 0.69 ± 0.01 | 0.69 ± 0.0 | 0.64 ± 0.01 | 0.67 ± 0.01 | 9.32 |

Table 13: Accuracy score for Mushrooms dataset for each shift compared for all the methods. A white color means the method does not increase the performance compared to Train Src (Train on the source). Green indicates that the performance improved with the DA methods. The darker the color, the more significant the change.

| | | enl→tap | tap→enl | Mean | Rank |
|---|---|---|---|---|---|
| | Train Src | 0.67 ± 0.01 | 0.77 ± 0.01 | 0.72 ± 0.01 | 8.50 |
| | Train Tgt | 1.0 ± 0.0 | 1.0 ± 0.0 | 1.0 ± 0.0 | 1.00 |
| Reweighting | Dens. RW | 0.67 ± 0.01 | 0.76 ± 0.0 | 0.71 ± 0.01 | 9.00 |
| | Disc. RW | 0.73 ± 0.06 | 0.78 ± 0.01 | 0.75 ± 0.04 | 4.50 |
| | Gauss. RW | 0.56 ± 0.0 | 0.46 ± 0.0 | 0.51 ± 0.0 | 17.50 |
| | KLIEP | 0.66 ± 0.02 | 0.77 ± 0.01 | 0.72 ± 0.01 | 10.75 |
| | KMM | 0.7 ± 0.02 | 0.78 ± 0.01 | 0.74 ± 0.01 | 6.00 |
| | NN RW | 0.67 ± 0.05 | 0.75 ± 0.01 | 0.71 ± 0.03 | 12.00 |
| | MMDTarS | 0.7 ± 0.02 | 0.77 ± 0.01 | 0.74 ± 0.01 | 6.00 |
| Mapping | CORAL | 0.66 ± 0.02 | 0.77 ± 0.01 | 0.72 ± 0.02 | 11.50 |
| | MapOT | 0.65 ± 0.01 | 0.62 ± 0.02 | 0.63 ± 0.01 | 14.00 |
| | EntOT | 0.82 ± 0.01 | 0.67 ± 0.01 | 0.75 ± 0.01 | 8.00 |
| | ClassRegOT | 0.63 ± 0.01 | 0.62 ± 0.01 | 0.62 ± 0.01 | 15.50 |
| | LinOT | 0.72 ± 0.01 | 0.81 ± 0.01 | 0.76 ± 0.01 | 4.00 |
| | MMD-LS | 0.85 ± 0.01 | NA | 0.85 ± 0.01 | 4.00 |
| Subspace | JPCA | 0.64 ± 0.02 | 0.78 ± 0.02 | 0.71 ± 0.02 | 10.00 |
| | SA | 0.53 ± 0.02 | 0.86 ± 0.01 | 0.7 ± 0.02 | 10.00 |
| Other | OTLabelProp | 0.68 ± 0.01 | 0.61 ± 0.01 | 0.64 ± 0.01 | 11.50 |

Table 14: Accuracy score for Phishing dataset for each shift compared for all the methods. A white color means the method does not increase the performance compared to Train Src (Train on the source). Green indicates that the performance improved with the DA methods. The darker the color, the more significant the change.

| | | ip_adress→no_ip_adress | no_ip_adress→ip_adress | Mean | Rank |
|---|---|---|---|---|---|
| | Train Src | 0.94 ± 0.01 | 0.88 ± 0.01 | 0.91 ± 0.01 | 7.0 |
| | Train Tgt | 0.97 ± 0.01 | 0.97 ± 0.01 | 0.97 ± 0.01 | 1.0 |
| Reweighting | Dens. RW | 0.94 ± 0.01 | 0.88 ± 0.01 | 0.91 ± 0.01 | 6.5 |
| | Disc. RW | 0.94 ± 0.01 | 0.88 ± 0.01 | 0.91 ± 0.01 | 9.5 |
| | Gauss. RW | 0.51 ± 0.0 | 0.41 ± 0.0 | 0.46 ± 0.0 | 20.0 |
| | KLIEP | 0.94 ± 0.01 | 0.89 ± 0.01 | 0.91 ± 0.01 | 5.0 |
| | KMM | 0.94 ± 0.01 | 0.89 ± 0.02 | 0.91 ± 0.01 | 6.5 |
| | NN RW | 0.94 ± 0.01 | 0.89 ± 0.01 | 0.91 ± 0.01 | 6.5 |
| | MMDTarS | 0.94 ± 0.01 | 0.88 ± 0.01 | 0.91 ± 0.01 | 5.5 |
| Mapping | CORAL | 0.93 ± 0.01 | 0.91 ± 0.01 | 0.92 ± 0.01 | 5.5 |
| | MapOT | 0.83 ± 0.01 | 0.84 ± 0.03 | 0.84 ± 0.02 | 15.0 |
| | EntOT | 0.87 ± 0.04 | 0.85 ± 0.03 | 0.86 ± 0.04 | 16.0 |
| | ClassRegOT | 0.87 ± 0.02 | 0.89 ± 0.02 | 0.88 ± 0.02 | 9.0 |
| | LinOT | 0.91 ± 0.01 | 0.9 ± 0.02 | 0.91 ± 0.02 | 7.0 |
| | MMD-LS | NA | 0.88 ± 0.01 | 0.88 ± 0.01 | 11.5 |
| Subspace | JPCA | 0.92 ± 0.01 | 0.89 ± 0.01 | 0.9 ± 0.01 | 8.0 |
| | SA | 0.9 ± 0.02 | 0.88 ± 0.02 | 0.89 ± 0.02 | 11.0 |
| | TSL | 0.88 ± 0.02 | 0.84 ± 0.02 | 0.86 ± 0.02 | 14.0 |
| Other | JDOT | 0.8 ± 0.02 | 0.8 ± 0.01 | 0.8 ± 0.02 | 18.0 |
| | OTLabelProp | 0.86 ± 0.01 | 0.86 ± 0.01 | 0.86 ± 0.01 | 16.0 |
| | DASVM | NA | 0.88 ± 0.01 | 0.88 ± 0.01 | 15.0 |

Table 15: Accuracy score for BCI dataset for each shift compared for all the methods. A white color means the method does not increase the performance compared to Train Src (Train on the source). Green indicates that the performance improved with the DA methods. The darker the color, the more significant the change.

| | | 1 | 2 | 3 | 4 | 5 | 6 | 7 | 8 | 9 | Mean | Rank |
|---|---|---|---|---|---|---|---|---|---|---|---|---|
| | Train Src | 0.59 ± 0.05 | 0.51 ± 0.05 | 0.72 ± 0.05 | 0.49 ± 0.06 | 0.39 ± 0.04 | 0.41 ± 0.06 | 0.62 ± 0.07 | 0.69 ± 0.08 | 0.54 ± 0.1 | 0.55 ± 0.06 | 8.56 |
| | Train Tgt | 0.74 ± 0.05 | 0.61 ± 0.07 | 0.82 ± 0.03 | 0.52 ± 0.04 | 0.41 ± 0.08 | 0.47 ± 0.06 | 0.75 ± 0.03 | 0.8 ± 0.03 | 0.61 ± 0.06 | 0.64 ± 0.05 | 1.44 |
| Reweighting | Dens. RW | 0.59 ± 0.06 | 0.51 ± 0.05 | 0.72 ± 0.04 | 0.5 ± 0.06 | 0.39 ± 0.04 | 0.4 ± 0.07 | 0.63 ± 0.06 | 0.67 ± 0.06 | 0.53 ± 0.09 | 0.55 ± 0.06 | 9.61 |
| | Disc. RW | 0.63 ± 0.06 | 0.51 ± 0.06 | 0.76 ± 0.05 | 0.49 ± 0.04 | 0.4 ± 0.04 | 0.42 ± 0.08 | 0.6 ± 0.06 | 0.68 ± 0.03 | 0.56 ± 0.09 | 0.56 ± 0.06 | 6.89 |
| | Gauss. RW | 0.28 ± 0.07 | 0.25 ± 0.01 | 0.25 ± 0.01 | 0.25 ± 0.01 | 0.25 ± 0.01 | 0.25 ± 0.01 | 0.25 ± 0.01 | 0.25 ± 0.01 | 0.25 ± 0.01 | 0.25 ± 0.01 | 20.44 |
| | KLIEP | 0.59 ± 0.05 | 0.52 ± 0.04 | 0.71 ± 0.04 | 0.48 ± 0.07 | 0.39 ± 0.04 | 0.41 ± 0.08 | 0.63 ± 0.07 | 0.69 ± 0.08 | 0.55 ± 0.08 | 0.55 ± 0.06 | 8.17 |
| | KMM | 0.61 ± 0.06 | 0.45 ± 0.08 | 0.74 ± 0.05 | 0.44 ± 0.04 | 0.31 ± 0.09 | 0.4 ± 0.07 | 0.64 ± 0.09 | 0.64 ± 0.01 | 0.52 ± 0.07 | 0.53 ± 0.06 | 10.89 |
| | NN RW | 0.58 ± 0.06 | 0.45 ± 0.05 | 0.69 ± 0.03 | 0.47 ± 0.09 | 0.36 ± 0.06 | 0.41 ± 0.05 | 0.63 ± 0.07 | 0.68 ± 0.07 | 0.54 ± 0.08 | 0.54 ± 0.06 | 10.67 |
| | MMDTarS | 0.59 ± 0.06 | 0.52 ± 0.04 | 0.73 ± 0.05 | 0.5 ± 0.06 | 0.39 ± 0.04 | 0.41 ± 0.06 | 0.62 ± 0.07 | 0.69 ± 0.07 | 0.54 ± 0.08 | 0.55 ± 0.06 | 8.11 |
| Mapping | CORAL | 0.77 ± 0.08 | 0.56 ± 0.06 | 0.81 ± 0.04 | 0.49 ± 0.08 | 0.45 ± 0.05 | 0.44 ± 0.06 | 0.73 ± 0.05 | 0.75 ± 0.1 | 0.56 ± 0.06 | 0.62 ± 0.07 | 2.67 |
| | MapOT | 0.61 ± 0.08 | 0.35 ± 0.04 | 0.64 ± 0.05 | 0.39 ± 0.03 | 0.32 ± 0.04 | 0.29 ± 0.05 | 0.53 ± 0.04 | 0.61 ± 0.08 | 0.49 ± 0.05 | 0.47 ± 0.05 | 11.11 |
| | EntOT | 0.67 ± 0.07 | 0.45 ± 0.05 | 0.79 ± 0.06 | 0.37 ± 0.04 | 0.33 ± 0.06 | 0.32 ± 0.02 | 0.67 ± 0.01 | 0.74 ± 0.07 | 0.52 ± 0.06 | 0.54 ± 0.05 | 9.39 |
| | ClassRegOT | 0.61 ± 0.07 | 0.43 ± 0.07 | 0.71 ± 0.07 | 0.4 ± 0.05 | 0.35 ± 0.04 | 0.31 ± 0.07 | 0.62 ± 0.06 | 0.7 ± 0.05 | 0.49 ± 0.06 | 0.51 ± 0.06 | 12.33 |
| | LinOT | 0.76 ± 0.1 | 0.55 ± 0.05 | 0.79 ± 0.04 | 0.48 ± 0.09 | 0.46 ± 0.06 | 0.46 ± 0.07 | 0.71 ± 0.04 | 0.76 ± 0.09 | 0.53 ± 0.07 | 0.61 ± 0.07 | 3.78 |
| | MMD-LS | 0.66 ± 0.1 | 0.46 ± 0.13 | 0.63 ± 0.16 | 0.35 ± 0.16 | 0.39 ± 0.05 | 0.34 ± 0.1 | 0.65 ± 0.08 | 0.66 ± 0.12 | 0.49 ± 0.08 | 0.51 ± 0.1 | 10.11 |
| Subspace | JPCA | 0.57 ± 0.07 | 0.39 ± 0.05 | 0.72 ± 0.06 | 0.47 ± 0.07 | 0.31 ± 0.05 | 0.33 ± 0.03 | 0.64 ± 0.06 | 0.68 ± 0.04 | 0.49 ± 0.09 | 0.51 ± 0.06 | 12.78 |
| | SA | 0.74 ± 0.1 | 0.58 ± 0.09 | 0.8 ± 0.03 | 0.48 ± 0.07 | 0.39 ± 0.04 | 0.4 ± 0.06 | 0.66 ± 0.1 | 0.73 ± 0.07 | 0.54 ± 0.06 | 0.59 ± 0.07 | 6.33 |
| | TCA | 0.31 ± 0.09 | 0.26 ± 0.03 | 0.24 ± 0.07 | 0.24 ± 0.1 | 0.27 ± 0.05 | 0.26 ± 0.07 | 0.27 ± 0.06 | 0.31 ± 0.1 | 0.29 ± 0.1 | 0.27 ± 0.07 | 19.50 |
| | TSL | 0.24 ± 0.04 | 0.26 ± 0.05 | 0.27 ± 0.05 | 0.26 ± 0.03 | 0.23 ± 0.07 | 0.27 ± 0.04 | 0.26 ± 0.01 | 0.24 ± 0.09 | 0.22 ± 0.03 | 0.25 ± 0.05 | 20.00 |
| Other | JDOT | 0.57 ± 0.07 | 0.37 ± 0.06 | 0.61 ± 0.05 | 0.39 ± 0.07 | 0.28 ± 0.04 | 0.3 ± 0.03 | 0.53 ± 0.09 | 0.62 ± 0.05 | 0.49 ± 0.05 | 0.46 ± 0.06 | 17.00 |
| | OTLabelProp | 0.63 ± 0.06 | 0.4 ± 0.06 | 0.64 ± 0.05 | 0.41 ± 0.04 | 0.35 ± 0.06 | 0.33 ± 0.06 | 0.59 ± 0.06 | 0.66 ± 0.04 | 0.5 ± 0.09 | 0.5 ± 0.06 | 13.11 |

Table 16: Accuracy score for BCI dataset for each shift compared for all the deep DA methods. A white color means the method does not increase the performance compared to Train Src (Train on the source). Green indicates that the performance improved with the DA methods. The darker the color, the more significant the change.

| | 1 | 2 | 3 | 4 | 5 | 6 | 7 | 8 | 9 | Mean | Rank |
|---|---|---|---|---|---|---|---|---|---|---|---|
| Train Src | 0.59 | 0.29 | 0.67 | 0.43 | 0.38 | 0.29 | 0.67 | 0.76 | 0.67 | 0.53 | 2.61 |
| Train Tgt | 0.57 | 0.43 | 0.71 | 0.53 | 0.34 | 0.29 | 0.62 | 0.69 | 0.79 | 0.55 | 2.17 |
| DANN | 0.52 | 0.24 | 0.55 | 0.38 | 0.4 | 0.22 | 0.38 | 0.66 | 0.64 | 0.44 | 4.44 |
| DeepCORAL | 0.55 | 0.26 | 0.66 | 0.36 | 0.48 | 0.33 | 0.57 | 0.72 | 0.66 | 0.51 | 2.72 |
| DeepJDOT | 0.53 | 0.34 | 0.64 | 0.43 | 0.43 | 0.43 | 0.57 | 0.62 | 0.6 | 0.51 | 3.17 |

## E.2   Impact of the base estimators on the simulated datasets

As mentioned in the main paper, it is possible to partly compensate for the shift by choosing the right base estimator. In this part, we provide the results on the Simulated dataset for three different base estimators: Logistic Regression (LR) in Table 17, SVM in Table 18, and XGBoost in Table 19. Observing the two first rows for covariate shift, we see that with LR (Table 17), there is a significant drop in performance between training on the source v.s. training on the target ($\sim 10\%$), while using SVC (Table 18) only leads to a drop ($\sim 3\%$). Finally, using XGBoost (Table 19) maintains the performance. The reweighting DA methods help compensate for the shift when using a simpler LR estimator. However when using an SVC, as shown in the main paper, the reweighting does not help to compensate for the covariate shift. If we look at the other shifts, the problem is harder. The subspace methods help with subspace shift, and the mapping methods help with the conditional shift.

These Tables show the importance of choosing the right base estimator. It is clear that choosing an appropriate base estimator can partially compensate for some shifts.

Table 17: Accuracy score for simulated datasets compared for all the methods with LR. A white color means the method does not increase the performance compared to Train Src (Train on the source). Green indicates that the performance improved with the DA methods. The darker the color, the more significant the change.

| | | Cov. shift | Tar. shift | Cond. shift | Sub. shift | Mean | Rank |
|---|---|---|---|---|---|---|---|
| | Train Src | 0.8 ± 0.02 | 0.81 ± 0.03 | 0.68 ± 0.03 | 0.06 ± 0.01 | 0.59 ± 0.02 | 10.50 |
| | Train Tgt | 0.91 ± 0.02 | 0.92 ± 0.01 | 0.79 ± 0.03 | 0.97 ± 0.01 | 0.9 ± 0.02 | 2.00 |
| Reweighting | Dens. RW | 0.88 ± 0.03 | 0.84 ± 0.04 | 0.66 ± 0.03 | 0.07 ± 0.02 | 0.61 ± 0.03 | 7.50 |
| | Disc. RW | 0.55 ± 0.02 | 0.78 ± 0.05 | 0.7 ± 0.04 | 0.06 ± 0.01 | 0.52 ± 0.03 | 13.25 |
| | Gauss. RW | 0.89 ± 0.02 | 0.85 ± 0.03 | 0.64 ± 0.03 | 0.06 ± 0.01 | 0.61 ± 0.02 | 8.00 |
| | KLIEP | 0.8 ± 0.02 | 0.81 ± 0.04 | 0.69 ± 0.03 | 0.07 ± 0.02 | 0.59 ± 0.03 | 8.25 |
| | KMM | 0.84 ± 0.03 | 0.82 ± 0.05 | 0.66 ± 0.04 | 0.07 ± 0.02 | 0.6 ± 0.04 | 7.88 |
| | NN RW | 0.81 ± 0.02 | 0.82 ± 0.04 | 0.67 ± 0.03 | 0.07 ± 0.01 | 0.59 ± 0.03 | 7.75 |
| | MMDTarS | 0.8 ± 0.02 | 0.84 ± 0.04 | 0.66 ± 0.03 | 0.07 ± 0.02 | 0.59 ± 0.03 | 10.75 |
| Mapping | CORAL | 0.73 ± 0.05 | 0.68 ± 0.11 | 0.75 ± 0.08 | 0.04 ± 0.02 | 0.55 ± 0.06 | 12.25 |
| | MapOT | 0.73 ± 0.03 | 0.6 ± 0.04 | 0.79 ± 0.03 | 0.03 ± 0.01 | 0.54 ± 0.03 | 13.75 |
| | EntOT | 0.72 ± 0.05 | 0.61 ± 0.04 | 0.79 ± 0.03 | 0.03 ± 0.01 | 0.54 ± 0.03 | 12.50 |
| | ClassRegOT | 0.87 ± 0.08 | 0.59 ± 0.04 | 0.79 ± 0.03 | 0.03 ± 0.01 | 0.57 ± 0.04 | 11.50 |
| | LinOT | 0.77 ± 0.03 | 0.65 ± 0.06 | 0.76 ± 0.04 | 0.04 ± 0.02 | 0.56 ± 0.04 | 12.00 |
| | MMD-LS | 0.7 ± 0.1 | 0.64 ± 0.06 | 0.78 ± 0.04 | 0.38 ± 0.22 | 0.63 ± 0.1 | 10.75 |
| Subspace | JPCA | 0.8 ± 0.02 | 0.81 ± 0.03 | 0.68 ± 0.03 | 0.06 ± 0.01 | 0.59 ± 0.02 | 11.25 |
| | SA | 0.8 ± 0.02 | 0.62 ± 0.04 | 0.78 ± 0.03 | 0.04 ± 0.02 | 0.56 ± 0.03 | 11.25 |
| | TCA | 0.44 ± 0.29 | 0.49 ± 0.06 | 0.54 ± 0.11 | 0.54 ± 0.23 | 0.5 ± 0.17 | 15.50 |
| | TSL | 0.8 ± 0.02 | 0.81 ± 0.03 | 0.68 ± 0.03 | 0.06 ± 0.01 | 0.59 ± 0.02 | 11.00 |
| Other | OTLabelProp | 0.73 ± 0.03 | 0.59 ± 0.04 | 0.79 ± 0.03 | 0.03 ± 0.01 | 0.53 ± 0.03 | 13.50 |

Table 18: Accuracy score for simulated datasets compared for all the methods with SVC. A white color means the method does not increase the performance compared to Train Src (Train on the source). Green indicates that the performance improved with the DA methods. The darker the color, the more significant the change.

| | | Cov. shift | Tar. shift | Cond. shift | Sub. shift | Mean | Rank |
|---|---|---|---|---|---|---|---|
| | Train Src | 0.88 ± 0.03 | 0.85 ± 0.04 | 0.66 ± 0.02 | 0.19 ± 0.03 | 0.65 ± 0.03 | 9.38 |
| | Train Tgt | 0.92 ± 0.02 | 0.93 ± 0.02 | 0.82 ± 0.03 | 0.98 ± 0.01 | 0.91 ± 0.02 | 1.25 |
| Reweighting | Dens. RW | 0.88 ± 0.03 | 0.86 ± 0.04 | 0.66 ± 0.02 | 0.18 ± 0.04 | 0.64 ± 0.03 | 8.88 |
| | Disc. RW | 0.85 ± 0.04 | 0.83 ± 0.04 | 0.72 ± 0.04 | 0.18 ± 0.03 | 0.64 ± 0.04 | 10.75 |
| | Gauss. RW | 0.89 ± 0.03 | 0.86 ± 0.04 | 0.65 ± 0.02 | 0.21 ± 0.04 | 0.65 ± 0.03 | 7.00 |
| | KLIEP | 0.88 ± 0.03 | 0.86 ± 0.04 | 0.66 ± 0.02 | 0.19 ± 0.03 | 0.65 ± 0.03 | 8.12 |
| | KMM | 0.89 ± 0.03 | 0.87 ± 0.04 | 0.64 ± 0.04 | 0.15 ± 0.05 | 0.64 ± 0.04 | 9.50 |
| | NN RW | 0.89 ± 0.03 | 0.86 ± 0.04 | 0.67 ± 0.02 | 0.15 ± 0.04 | 0.64 ± 0.03 | 9.12 |
| | MMDTarS | 0.88 ± 0.03 | 0.86 ± 0.04 | 0.64 ± 0.03 | 0.2 ± 0.04 | 0.65 ± 0.03 | 9.12 |
| Mapping | CORAL | 0.74 ± 0.04 | 0.7 ± 0.11 | 0.76 ± 0.08 | 0.18 ± 0.04 | 0.59 ± 0.07 | 11.50 |
| | MapOT | 0.72 ± 0.04 | 0.57 ± 0.04 | 0.82 ± 0.03 | 0.02 ± 0.01 | 0.53 ± 0.03 | 14.25 |
| | EntOT | 0.71 ± 0.04 | 0.6 ± 0.04 | 0.82 ± 0.03 | 0.12 ± 0.06 | 0.56 ± 0.05 | 12.75 |
| | ClassRegOT | 0.74 ± 0.09 | 0.58 ± 0.04 | 0.81 ± 0.03 | 0.11 ± 0.06 | 0.56 ± 0.06 | 12.75 |
| | LinOT | 0.73 ± 0.05 | 0.73 ± 0.08 | 0.76 ± 0.06 | 0.18 ± 0.04 | 0.6 ± 0.06 | 11.75 |
| | MMD-LS | 0.65 ± 0.08 | 0.68 ± 0.11 | 0.79 ± 0.05 | 0.55 ± 0.31 | 0.67 ± 0.14 | 10.75 |
| Subspace | JPCA | 0.88 ± 0.03 | 0.85 ± 0.04 | 0.66 ± 0.02 | 0.15 ± 0.05 | 0.64 ± 0.04 | 11.25 |
| | SA | 0.74 ± 0.04 | 0.68 ± 0.04 | 0.8 ± 0.03 | 0.11 ± 0.03 | 0.58 ± 0.03 | 12.50 |
| | TCA | 0.46 ± 0.21 | 0.48 ± 0.09 | 0.55 ± 0.11 | 0.56 ± 0.2 | 0.51 ± 0.15 | 15.62 |
| | TSL | 0.88 ± 0.03 | 0.85 ± 0.04 | 0.66 ± 0.02 | 0.19 ± 0.03 | 0.65 ± 0.03 | 9.62 |
| Other | OTLabelProp | 0.72 ± 0.04 | 0.58 ± 0.04 | 0.81 ± 0.04 | 0.04 ± 0.05 | 0.54 ± 0.04 | 14.00 |

Table 19: Accuracy score for simulated datasets compared for all the methods with XGBoost. A white color means the method does not increase the performance compared to Train Src (Train on the source). Green indicates that the performance improved with the DA methods. The darker the color, the more significant the change.

| | | Cov. shift | Tar. shift | Cond. shift | Sub. shift | Mean | Rank |
|---|---|---|---|---|---|---|---|
| | Train Src | 0.89 ± 0.02 | 0.84 ± 0.04 | 0.66 ± 0.03 | 0.21 ± 0.03 | 0.65 ± 0.03 | 9.25 |
| | Train Tgt | 0.89 ± 0.02 | 0.93 ± 0.02 | 0.77 ± 0.03 | 0.98 ± 0.01 | 0.89 ± 0.02 | 2.25 |
| Reweighting | Dens. RW | 0.88 ± 0.03 | 0.84 ± 0.03 | 0.67 ± 0.03 | 0.22 ± 0.04 | 0.65 ± 0.03 | 8.25 |
| | Disc. RW | 0.68 ± 0.06 | 0.84 ± 0.03 | 0.66 ± 0.03 | 0.2 ± 0.03 | 0.6 ± 0.04 | 12.25 |
| | Gauss. RW | 0.87 ± 0.03 | 0.84 ± 0.03 | 0.67 ± 0.03 | 0.22 ± 0.03 | 0.65 ± 0.03 | 9.12 |
| | KLIEP | 0.88 ± 0.03 | 0.84 ± 0.03 | 0.67 ± 0.03 | 0.21 ± 0.03 | 0.65 ± 0.03 | 7.12 |
| | KMM | 0.87 ± 0.04 | 0.84 ± 0.04 | 0.67 ± 0.04 | 0.22 ± 0.04 | 0.65 ± 0.04 | 7.62 |
| | NN RW | 0.88 ± 0.03 | 0.84 ± 0.04 | 0.66 ± 0.03 | 0.2 ± 0.03 | 0.65 ± 0.03 | 10.50 |
| | MMDTarS | 0.88 ± 0.03 | 0.86 ± 0.04 | 0.63 ± 0.03 | 0.22 ± 0.03 | 0.65 ± 0.03 | 7.50 |
| Mapping | CORAL | 0.71 ± 0.04 | 0.71 ± 0.11 | 0.74 ± 0.08 | 0.17 ± 0.05 | 0.58 ± 0.07 | 12.75 |
| | MapOT | 0.7 ± 0.04 | 0.59 ± 0.03 | 0.8 ± 0.03 | 0.17 ± 0.05 | 0.56 ± 0.04 | 13.25 |
| | EntOT | 0.69 ± 0.05 | 0.61 ± 0.04 | 0.8 ± 0.03 | 0.2 ± 0.02 | 0.57 ± 0.04 | 12.25 |
| | ClassRegOT | 0.82 ± 0.11 | 0.59 ± 0.03 | 0.8 ± 0.03 | 0.16 ± 0.04 | 0.59 ± 0.05 | 12.00 |
| | LinOT | 0.72 ± 0.04 | 0.68 ± 0.06 | 0.76 ± 0.04 | 0.19 ± 0.04 | 0.59 ± 0.05 | 12.00 |
| | MMD-LS | 0.64 ± 0.07 | 0.68 ± 0.08 | 0.78 ± 0.04 | 0.59 ± 0.25 | 0.67 ± 0.11 | 10.25 |
| Subspace | JPCA | 0.88 ± 0.03 | 0.84 ± 0.03 | 0.67 ± 0.03 | 0.14 ± 0.05 | 0.63 ± 0.03 | 10.50 |
| | SA | 0.72 ± 0.04 | 0.69 ± 0.04 | 0.78 ± 0.03 | 0.13 ± 0.04 | 0.58 ± 0.04 | 11.75 |
| | TCA | 0.48 ± 0.05 | 0.5 ± 0.05 | 0.51 ± 0.05 | 0.51 ± 0.06 | 0.5 ± 0.05 | 15.50 |
| | TSL | 0.89 ± 0.02 | 0.84 ± 0.04 | 0.66 ± 0.03 | 0.21 ± 0.03 | 0.65 ± 0.03 | 9.25 |
| Other | OTLabelProp | 0.72 ± 0.05 | 0.59 ± 0.04 | 0.81 ± 0.04 | 0.04 ± 0.05 | 0.54 ± 0.04 | 13.00 |

### E.3 Unrealistic validation with supervised scorer

Table 20 shows the results when we choose the supervised scorer that is when validating on target labels. It is important to highlight that this choice is impossible in real life applications due to the lack of target labels. When using the target labels, the method's parameters are better validated. This can be seen by the significant increase in the table (blue values), which are numerous in this table compared to the one with the selected realistic scorer. For example, the method MMDTarS, which is made for Target shift, compensates all the shift simulated covariate shifts when we select the model with a supervised scorer. When looking at the rank, 11 DA methods have a higher rank than Train Src compared to 9 when using realistic scorer. The findings hold for Deep DA where the accuracy in Table 21 is overall better than when using unsupervised scorers.

Table 20: Accuracy score for all datasets compared for all the methods for simulated and real-life datasets. In this table, each DA method is validated with the supervised scorer. The color indicates the amount of the improvement. A white color means the method is not statistically different from Train Src (Train on source). Blue indicates that the performance improved with the DA methods, while red indicates a decrease. The darker the color, the more significant the change.

|  |  | Cov. shift | Tar. shift | Cond. shift | Sub. shift | Office31 | OfficeHome | MNIST/USPS | 20NewsGroups | AmazonReview | Mushrooms | Phishing | BCI | Rank |
|---|---|---|---|---|---|---|---|---|---|---|---|---|---|---|
|  | Train Src | 0.88 | 0.85 | 0.66 | 0.19 | 0.65 | 0.56 | 0.54 | 0.59 | 0.7 | 0.72 | 0.91 | 0.55 | 10.66 |
|  | Train Tgt | 0.92 | 0.93 | 0.82 | 0.98 | 0.89 | 0.8 | 0.96 | 1.0 | 0.73 | 1.0 | 0.97 | 0.64 | 1.55 |
| Reweighting | Dens. RW | 0.89 | 0.87 | 0.67 | 0.2 | 0.65 | 0.56 | 0.54 | 0.59 | 0.7 | 0.76 | 0.91 | 0.55 | 12.20 |
| | Disc. RW | 0.86 | 0.84 | 0.73 | 0.23 | 0.64 | 0.54 | 0.54 | 0.62 | 0.69 | 0.78 | 0.91 | 0.56 | 8.75 |
| | Gauss. RW | 0.89 | 0.86 | 0.65 | 0.21 | 0.22 | 0.44 | 0.11 | 0.54 | 0.55 | 0.51 | 0.46 | 0.25 | 16.45 |
| | KLIEP | 0.89 | 0.88 | 0.66 | 0.2 | 0.65 | 0.56 | 0.54 | 0.58 | 0.7 | 0.75 | 0.92 | 0.55 | 10.56 |
| | KMM | 0.89 | 0.87 | 0.67 | 0.19 | 0.64 | 0.55 | 0.53 | 0.71 | 0.66 | 0.75 | 0.92 | 0.54 | 11.74 |
| | NN RW | 0.89 | 0.86 | 0.67 | 0.15 | 0.65 | 0.55 | 0.55 | 0.59 | 0.66 | 0.72 | 0.91 | 0.54 | 9.15 |
| | MMDTarS | 0.88 | 0.93 | 0.66 | 0.27 | 0.65 | 0.56 | 0.54 | 0.59 | 0.7 | 0.74 | 0.91 | 0.56 | 10.81 |
| Mapping | CORAL | 0.74 | 0.84 | 0.82 | 0.19 | 0.66 | 0.57 | 0.62 | 0.75 | 0.7 | 0.72 | 0.92 | 0.62 | 5.08 |
| | MapOT | 0.87 | 0.63 | 0.82 | 0.14 | 0.6 | 0.51 | 0.6 | 0.77 | 0.68 | 0.63 | 0.84 | 0.47 | 10.21 |
| | EntOT | 0.89 | 0.61 | 0.82 | 0.47 | 0.66 | 0.58 | 0.63 | 0.88 | 0.68 | 0.81 | 0.87 | 0.53 | 9.40 |
| | ClassRegOT | 0.91 | 0.59 | 0.82 | 0.15 | NA | 0.59 | 0.66 | 0.98 | 0.68 | 0.89 | 0.9 | 0.52 | 8.25 |
| | LinOT | 0.89 | 0.81 | 0.81 | 0.19 | 0.66 | 0.58 | 0.65 | 0.88 | 0.71 | 0.81 | 0.91 | 0.61 | 4.06 |
| | MMD-LS | 0.88 | 0.85 | 0.81 | 0.73 | 0.65 | 0.56 | 0.56 | 0.98 | 0.69 | 0.89 | NA | 0.58 | 8.22 |
| Subspace | JPCA | 0.88 | 0.85 | 0.66 | 0.19 | 0.65 | 0.56 | 0.56 | 0.84 | 0.7 | 0.8 | 0.9 | 0.55 | 8.98 |
| | SA | 0.74 | 0.81 | 0.8 | 0.13 | 0.66 | 0.57 | 0.56 | 0.93 | 0.7 | 0.91 | 0.89 | 0.59 | 7.80 |
| | TCA | 0.4 | 0.46 | 0.5 | 0.58 | 0.04 | 0.02 | 0.11 | 0.49 | 0.61 | 0.46 | 0.49 | 0.27 | 17.58 |
| | TSL | 0.88 | 0.85 | 0.66 | 0.86 | 0.62 | 0.48 | 0.45 | 0.7 | 0.69 | 0.57 | 0.9 | 0.26 | 15.09 |
| Other | JDOT | 0.72 | 0.57 | 0.82 | 0.13 | 0.61 | 0.41 | 0.57 | 0.8 | 0.67 | 0.63 | 0.8 | 0.46 | 11.42 |
| | OTLabelProp | 0.9 | 0.76 | 0.81 | 0.14 | 0.66 | 0.56 | 0.64 | 0.89 | 0.67 | 0.69 | 0.86 | 0.51 | 10.01 |
| | DASVM | 0.89 | 0.86 | 0.64 | 0.12 | NA | NA | NA | 0.83 | NA | 0.76 | 0.86 | NA | 7.29 |

Table 21: Accuracy score compared for the Deep methods with the supervised scorer for a selection of real-life datasets.

| | MNIST/USPS | Office31 | OfficeHome | BCI | Rank |
|---|---|---|---|---|---|
| Train Src | 0.85 | 0.77 | 0.58 | 0.54 | 6.19 |
| Train Tgt | 0.98 | 0.96 | 0.83 | 0.56 | 2.07 |
| DeepCORAL | 0.93 | 0.82 | 0.63 | 0.54 | 3.29 |
| DAN | 0.91 | 0.79 | 0.61 | 0.55 | 4.76 |
| DANN | 0.9 | 0.76 | 0.6 | 0.42 | 4.98 |
| DeepJDOT | 0.93 | 0.83 | 0.63 | 0.54 | 2.92 |
| MCC | 0.94 | 0.81 | 0.66 | 0.55 | 2.38 |
| MDD | 0.91 | 0.83 | 0.58 | 0.42 | 4.96 |
| SPA | 0.92 | 0.78 | 0.56 | 0.4 | 5.39 |

Table 22: F1 score for all datasets compared for all the shallow methods for simulated and real-life datasets. The color indicates the amount of the improvement. A white color means the method is not statistically different from Train Src (Train on source). Blue indicates that the score improved with the DA methods, while red indicates a decrease. The darker the color, the more significant the change.

| | | Cov. shift | Tar. shift | Cond. shift | Sub. shift | Office31 | OfficeHome | MNIST/USPS | 20NewsGroups | AmazonReview | Mushrooms | Phishing | BCI | Selected Scorer | Rank |
|---|---|---|---|---|---|---|---|---|---|---|---|---|---|---|---|
| | Train Src | 0.88 | 0.87 | 0.62 | 0.17 | 0.64 | 0.56 | 0.52 | 0.56 | 0.65 | 0.72 | 0.91 | 0.53 | | 10.95 |
| | Train Tgt | 0.92 | 0.92 | 0.82 | 0.98 | 0.89 | 0.8 | 0.96 | 1.0 | 0.69 | 1.0 | 0.97 | 0.63 | | 1.70 |
| Reweighting | Dens. RW | 0.88 | 0.88 | 0.62 | 0.16 | 0.61 | 0.56 | 0.52 | 0.55 | 0.65 | 0.72 | 0.91 | 0.52 | IW | 12.71 |
| | Disc. RW | 0.85 | 0.86 | 0.7 | 0.16 | 0.63 | 0.53 | 0.48 | 0.56 | 0.63 | 0.76 | 0.91 | 0.54 | CircV | 8.39 |
| | Gauss. RW | 0.89 | 0.88 | 0.61 | 0.18 | 0.15 | 0.4 | 0.03 | 0.43 | 0.49 | 0.35 | 0.29 | 0.1 | CircV | 16.53 |
| | KLIEP | 0.88 | 0.88 | 0.62 | 0.17 | 0.64 | 0.55 | 0.52 | 0.56 | 0.65 | 0.72 | 0.91 | 0.52 | IW | 10.66 |
| | KMM | 0.89 | 0.87 | 0.6 | 0.15 | 0.63 | 0.54 | 0.51 | 0.69 | 0.5 | 0.74 | 0.91 | 0.49 | CircV | 11.58 |
| | NN RW | 0.89 | 0.88 | 0.63 | 0.14 | 0.64 | 0.55 | 0.52 | 0.56 | 0.64 | 0.71 | 0.91 | 0.5 | CircV | 8.22 |
| | MMDTarS | 0.88 | 0.88 | 0.6 | 0.17 | 0.57 | 0.56 | 0.52 | 0.56 | 0.65 | 0.74 | 0.91 | 0.53 | IW | 11.02 |
| Mapping | CORAL | 0.74 | 0.76 | 0.74 | 0.16 | 0.65 | 0.57 | 0.62 | 0.72 | 0.65 | 0.72 | 0.92 | 0.62 | CircV | 5.00 |
| | MapOT | 0.72 | 0.65 | 0.82 | 0.02 | 0.59 | 0.5 | 0.61 | 0.76 | 0.59 | 0.63 | 0.84 | 0.47 | PE | 10.49 |
| | EntOT | 0.71 | 0.67 | 0.82 | 0.12 | 0.63 | 0.57 | 0.59 | 0.83 | 0.49 | 0.75 | 0.85 | 0.53 | CircV | 10.15 |
| | ClassRegOT | 0.74 | 0.66 | 0.81 | 0.11 | NA | 0.53 | 0.62 | 0.97 | 0.67 | 0.82 | 0.89 | 0.52 | IW | 6.49 |
| | LinOT | 0.73 | 0.78 | 0.75 | 0.16 | 0.65 | 0.57 | 0.64 | 0.81 | 0.65 | 0.76 | 0.91 | 0.61 | CircV | 4.20 |
| | MMD-LS | 0.77 | 0.77 | 0.75 | 0.54 | 0.64 | 0.56 | 0.54 | 0.97 | 0.6 | 0.85 | NA | 0.48 | MixVal | 7.58 |
| Subspace | JPCA | 0.88 | 0.87 | 0.62 | 0.14 | 0.61 | 0.47 | 0.5 | 0.76 | 0.61 | 0.78 | 0.9 | 0.51 | PE | 9.55 |
| | SA | 0.73 | 0.74 | 0.8 | 0.1 | 0.64 | 0.57 | 0.55 | 0.88 | 0.56 | 0.77 | 0.89 | 0.52 | CircV | 7.95 |
| | TCA | 0.51 | 0.56 | 0.5 | 0.61 | 0.0 | 0.0 | 0.02 | 0.53 | 0.46 | 0.44 | 0.47 | 0.19 | DEV | 17.94 |
| | TSL | 0.88 | 0.87 | 0.63 | 0.17 | 0.63 | 0.47 | 0.45 | 0.59 | 0.58 | 0.28 | 0.9 | 0.21 | PE | 15.46 |
| Other | JDOT | 0.72 | 0.66 | 0.82 | 0.13 | 0.59 | 0.41 | 0.59 | 0.8 | 0.61 | 0.65 | 0.79 | 0.46 | IW | 10.74 |
| | OTLabelProp | 0.72 | 0.67 | 0.8 | 0.07 | 0.65 | 0.54 | 0.62 | 0.86 | 0.58 | 0.64 | 0.86 | 0.49 | CircV | 10.80 |
| | DASVM | 0.89 | 0.88 | 0.61 | 0.13 | NA | NA | NA | 0.87 | NA | 0.82 | 0.85 | NA | MixVal | 7.12 |

## E.4  F1-score of benchmark

We provide in Table 22 a version of Table 2 where the performance measure reported is the F1-score. It is interesting to note that the dynamic of which methods work best and are the more robust is very similar to the accuracy performance which illustrate the robustness of the benchmark.

## E.5  Comparison of the rank of DA scorer

To provide a more detailed assessment of the scorers' performance, we present a critical difference diagram of their rankings in Figure 4. The diagram highlights that the unrealistic supervised scorer significantly outperforms all others. Among the unsupervised scorers, CircV and IW achieve the best performance, with their rankings being very close and not statistically different according to a statistical test. Next, we observe a group comprising PE, DEV, and MixVal, where DEV and MixVal are also not statistically distinguishable. Finally, SND emerges as the worst-performing scorer in the benchmark.

To give a more detailed perspective, we present a visualization in Figure 5, showing the rank of each scorer for each DA method. In the right part of the figure, the supervised scorer (in pink) is consistently the top-ranked, as expected, across all methods. Similarly, CircV (in red) and IW (in orange) consistently outperform other scorers.

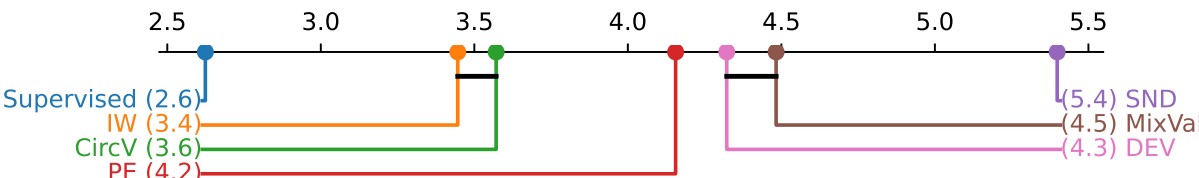

Figure 4: Critical difference diagram of average ranks for scorers, computed across shallow methods and shifts (lower ranks indicate better performance). Black lines between scorers indicate pairs that are not statistically different based on the Wilcoxon test.

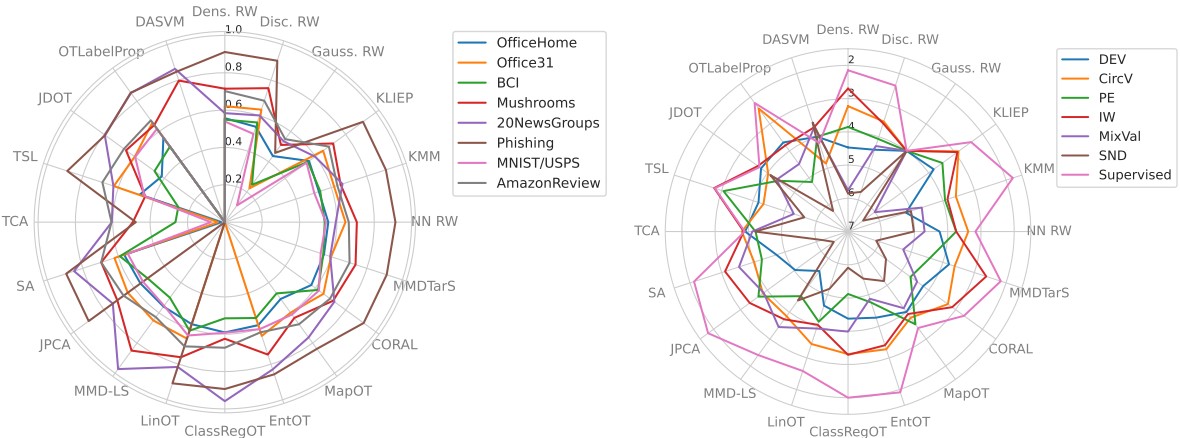

Figure 5: Illustrations as spider plots for all methods of the accuracy on each dataset (left) and the scorers rankings (right). For methods with no accuracy results (NA in Table 2) we replace the value by 0. We provide both spider plot in the same Figure to allow a comparison of the scorer ranking while having the possibility to check the performance for each method.

### E.6 Comparisons between supervised and unsupervised scorers

**Impact on the cross-validation score.** We observe in Figure 7 the cross-validation score as a function of the final accuracy for various DA methods type and for both supervised and unsupervised scorers. As expected, we observe a good correlation between accuracy and cross-validation score with the supervised scorer. An important remark is that the Circular Validation (CircV) (Bruzzone & Marconcini, 2010b) shows some correlation between accuracy and cross-validation score. It indicates that this unsupervised scorer might be the most suitable choice for hyperparameter selection. Using spearman correlation shows the same conclusion (see Figure 6). This is supported by our extended experimental results in Table 2 for which the CircV is selected as the best scorer the most often. A similar trend can be observed for the Importance Weighted (IW) (Sugiyama et al., 2007a) which is also confirmed in Table 2.

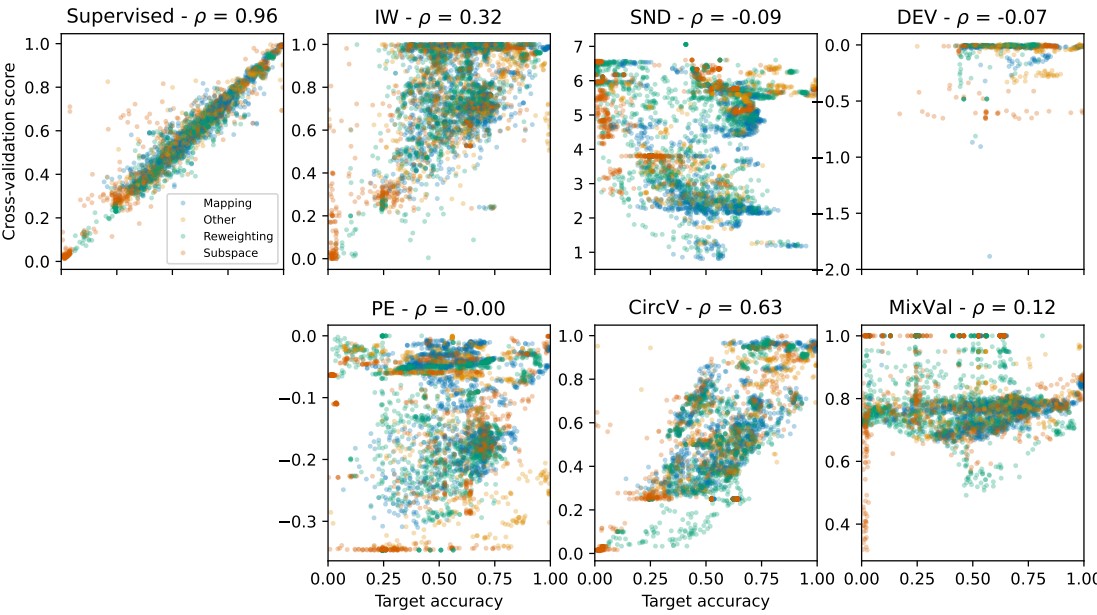

Figure 6: Cross-val score as a function of the accuracy for different supervised and unsupervised scorers. The Spearman correlation coefficient is reported for each scorer by $\rho$. Each point represents an inner split with a DA method (color of the points) and a dataset. A good score should correlate with the target accuracy.

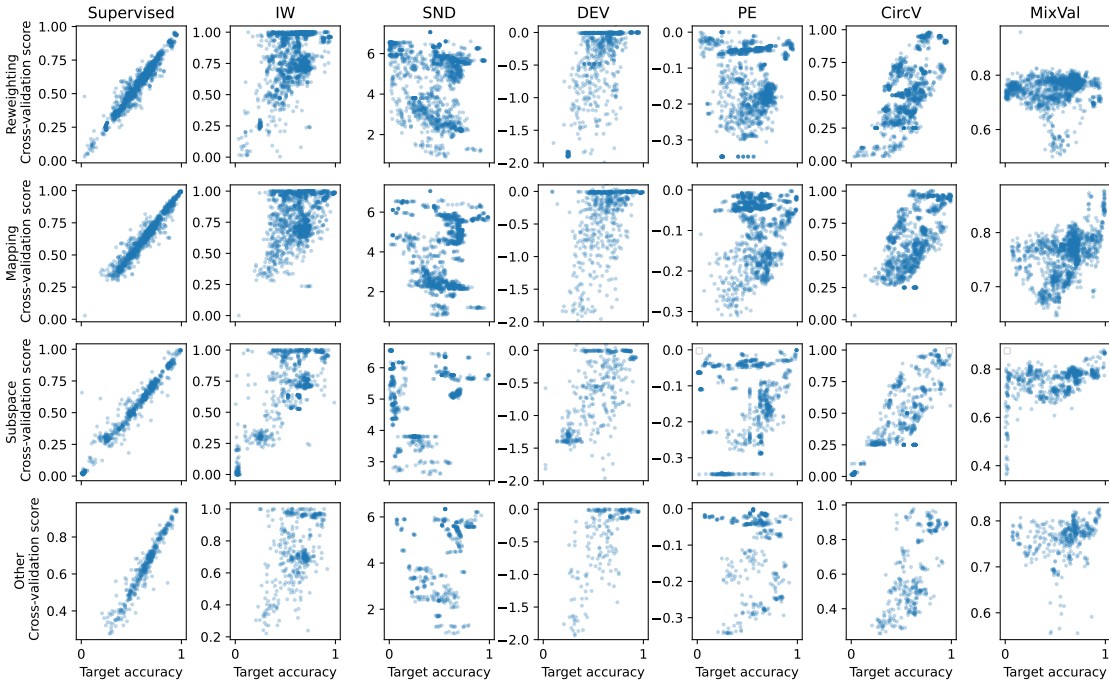

Figure 7: Cross-val score as a function of the accuracy for various DA methods and different supervised and unsupervised scorers. Each point represents an inner split with a DA method (color of the points) and a dataset. A good scorer should have a score that correlates with the target accuracy.

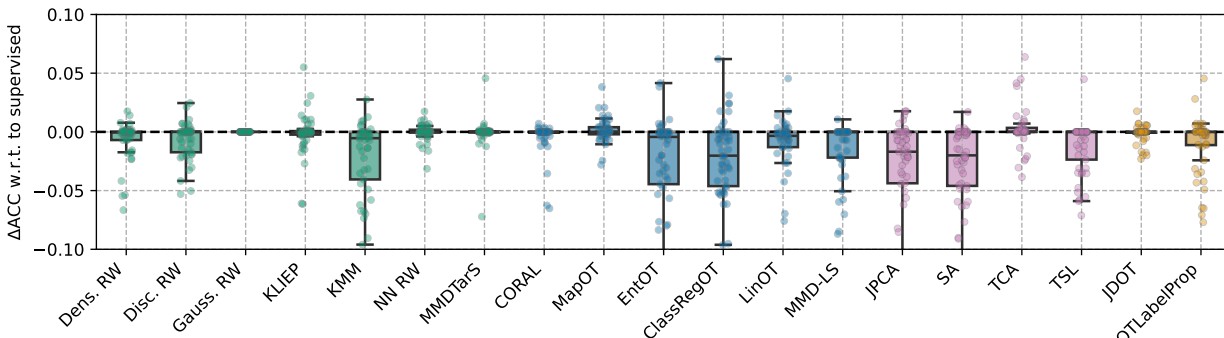

Figure 8: Change of accuracy of the DA methods with the best realistic unsupervised scorer (Table 2) w.r.t. the supervised scorer.

**Supervised scorer v.s. the best realistic unsupervised scorer.** We plot the loss in performance of the DA methods with the best realistic unsupervised scorer compared to using the supervised scorers in Figure 8.

**Supervised scorer v.s. realistic unsupervised scorers.** We present a scatter plot in Figure 9 and Figure 10 , the accuracy of different DA methods using both supervised scorer and unsupervised scorer. In this figure, points below the diagonal indicate a decrease in performance when using the unsupervised scorer compared to the supervised one. The colors represent different types of DA methods. We can see that the SND, DEV and PE scorers all lead to a large performance loss compared to the supervised scorer. While IW and CircV results are much more concentrated near the diagonal, indicating a small loss in performance. This concentration explains why these two scorers have been selected as the best scorers for most of the methods in Table 2.

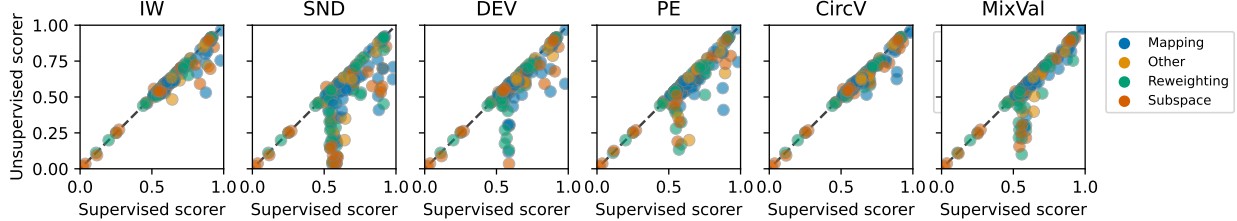

Figure 9: Accuracy of the DA methods using unsupervised scorers as a function of the accuracy with the supervised scorer. Colors represent the type of DA methods.

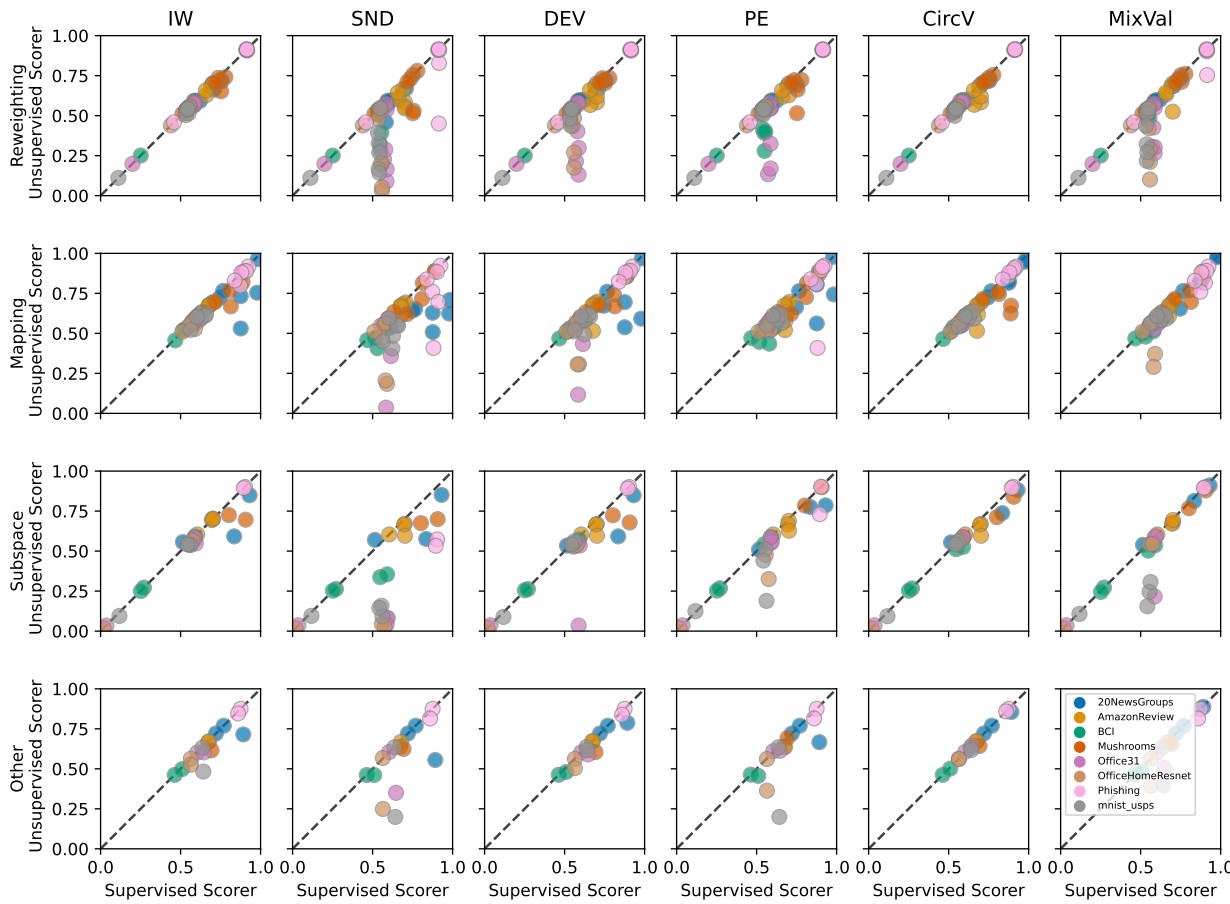

Figure 10: Accuracy of the DA methods using unsupervised scorers as a function of the accuracy with the supervised scorer for the different types of DA methods. Points below the diagonal represent a decrease in performance when using the unsupervised scorer compared to the supervised one. Colors represent the dataset on which the DA method is applied.

### E.7   Computational efficiency of the DA methods

Figure 11 shows the average computation time for training and testing each method. These results are based on one outer split, while we ran the benchmarks for five outer splits. Each method has a different time complexity. Interestingly, more time-consuming methods are not necessarily more performant than others. For instance, the highest-ranked methods—LinOT, CORAL, and SA—also have some of the lowest training and testing times. It's also worth noting that during the experiments, we enforced a 4-hour timeout. Thus, the more time-intensive methods might have been even slower without this timeout.

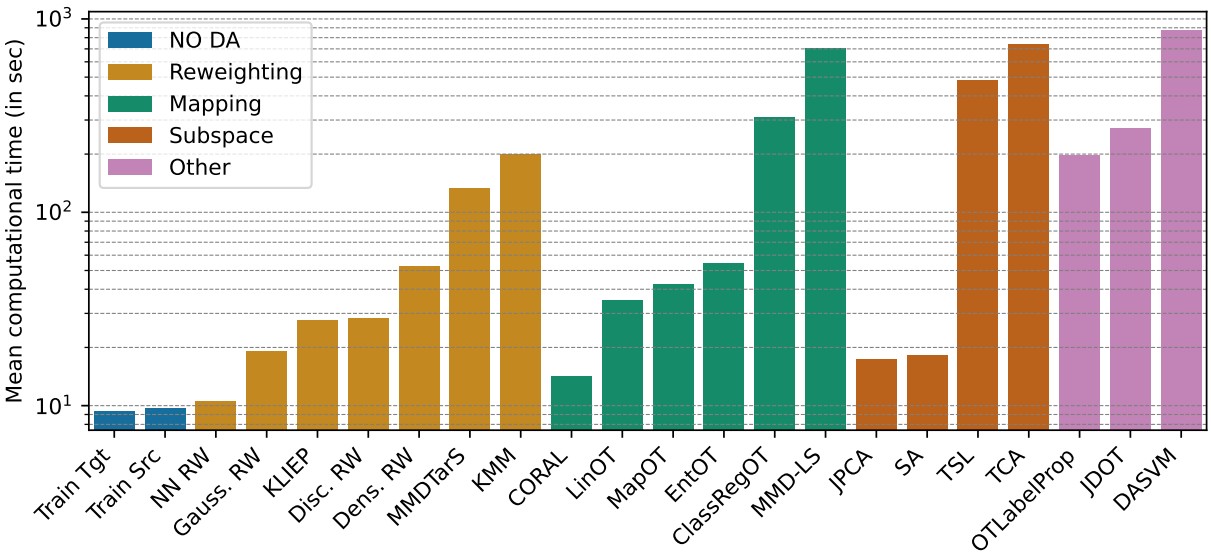

Figure 11: Mean computing time to train and test each method for every experiment outer split.

## E.8 Hyperparameters grid search for the DA methods and neural networks training

In this section, we first report the grids of hyperparameters used in our grid search for each DA method.

We also detail the configuration and hyperparameter grids for training neural networks in our Deep DA benchmark. We provide an overview of the key settings for each dataset, including batch sizes, optimizer parameters, learning rates, and the number of training epochs. Additionally, we outline the hyperparameter grids used for grid search across the Deep DA methods.

Table 23: Hyperparameter grids used in the grid search for each DA method. The hyperparameter grids were designed to be minimal yet expressive, allowing each method to perform optimally. We selected parameters based on what seemed most reasonable, according to our best knowledge.

| Method | Hyperparameter Grid |
|---|---|
| KLIEP | 'cv': [5],
'gamma': [0.0001, 0.001, 0.01, 0.1, 1.0, 10.0, 100.0, 1000.0, 'auto', 'scale'],
'max_iter': [1000],
'n_centers': [100],
'random_state': [0],
'tol': [1e-06] |
| KMM | 'B': [1000.0],
'gamma': [0.0001, 0.001, 0.01, 0.1, 1.0, 10.0, 100.0, 1000.0, None],
'max_iter': [1000],
'smooth_weights': [False],
'tol': [1e-06] |
| NN RW | 'laplace_smoothing': [True, False] |
| MapOT | 'max_iter': [1000000],
'metric': ['sqeuclidean', 'cosine', 'cityblock'],
'norm': ['median'] |
| JPCA | 'n_components': [1, 2, 5, 10, 20, 50, 100] |
| SA | 'n_components': [1, 2, 5, 10, 20, 50, 100] |
| TCA | 'kernel': ['rbf'],
'mu': [10, 100],
'n_components': [1, 2, 5, 10, 20, 50, 100] |
| CORAL | 'assume_centered': [False, True],
'reg': ['auto'] |
| MMDTarS | 'gamma': [0.0001, 0.001, 0.01, 0.1, 1.0, 10.0, 100.0, 1000.0, None],
'max_iter': [1000],
'reg': [1e-06],
'tol': [1e-06] |
| ClassRegOT | 'max_inner_iter': [1000],
'max_iter': [10],
'metric': ['sqeuclidean', 'cosine', 'cityblock'],
'norm': ['lpl1'],
'tol': [1e-06],
'(reg_cl, reg_e)': [([0.1], [0.1]), ([0.5], [0.5]), ([1.0], [1.0])] |
| Dens. RW | 'bandwidth': [0.01, 0.1, 1.0, 10.0, 100.0, 'scott', 'silverman'] |
| Disc. RW | 'domain_classifier': ['LR', 'SVC', 'XGB'] |
| Gauss. RW | 'reg': ['auto'] |
| DASVM | 'max_iter': [200] |
| JDOT | 'alpha': [0.1, 0.3, 0.5, 0.7, 0.9],
'n_iter_max': [100],
'thr_weights': [1e-07],
'tol': [1e-06] |
| EntOT | 'max_iter': [1000],
'metric': ['sqeuclidean', 'cosine', 'cityblock'],
'norm': ['median'],
'reg_e': [0.1, 0.5, 1.0],
'tol': [1e-06] |
| LinOT | 'bias': [True, False],
'reg': [1e-08, 1e-06, 0.1, 1, 10] |
| TSL | 'base_method': ['flda'],
'length_scale': [2],
'max_iter': [300],
'mu': [0.1, 1, 10],
'n_components': [1, 2, 5, 10, 20, 50, 100],
'reg': [0.0001],
'tol': [0.0001] |
| MMD-LS | 'gamma': [0.01, 0.1, 1, 10, 100],
'max_iter': [20],
'reg_k': [1e-08],
'reg_m': [1e-08],
'tol': [1e-05] |
| OTLabelProp | 'metric': ['sqeuclidean', 'cosine', 'cityblock'],
'(n_iter_max, reg)': [([10000], [None]), ([100], [0.1, 1])] |

Table 24: Configuration of Deep learning models for each dataset. This includes recommended Batch sizes, optimizer settings, learning rates, and maximum epochs.

| Dataset | Configuration |
|---|---|
| `mnist_usps` | <ul><li>**Neural net**: 2-layer CNN</li><li>**Batch size**: 256</li><li>**Optimizer**: SGD, momentum=0.6, weight_decay=1e-5</li><li>**Learning rate**: 0.1</li><li>**Epochs**: 20</li><li>**Learning rate scheduler**: LRScheduler(StepLR, step_size=10, gamma=0.2)</li></ul> |
| `office31` | <ul><li>**Neural net**: ResNet50</li><li>**Batch size**: 128</li><li>**Optimizer**: SGD, momentum=0.2, weight_decay=1e-5</li><li>**Learning rate**: 0.5</li><li>**Epochs**: 30</li><li>**Learning rate scheduler**: StepLR, step_size=10, gamma=0.2</li></ul> |
| `officehome` | <ul><li>**Neural net**: ResNet50</li><li>**Batch size**: 128</li><li>**Optimizer**: SGD, momentum=0.6, weight_decay=1e-5</li><li>**Learning rate**: 0.05</li><li>**Epochs**: 20</li><li>**Learning rate scheduler**: StepLR, step_size=10, gamma=0.2</li></ul> |
| `bci` | <ul><li>**Neural net**: FBCSPNet</li><li>**Batch size**: 64</li><li>**Optimizer**: AdamW</li><li>**Learning rate**: 0.000625</li><li>**Epochs**: 200</li><li>**Learning rate scheduler**: CosineAnnealingLR</li></ul> |

Table 25: Hyperparameter grids used in the grid search for each Deep DA method. The hyperparameter grids were designed to be minimal yet expressive, allowing each method to perform optimally. We selected parameters based on what seemed most reasonable, according to our best knowledge.

| Method | Hyperparameter Grid |
|---|---|
| DANN | 'reg': [0.001, 0.01, 0.1, 1.0] |
| DeepCORAL | 'reg': [0.00001, 0.0001, 0.001, 0.01, 0.1, 1.0, 10.0, 100.0, 1000.0] |
| DeepJDOT | 'reg_cl': [0.0001, 0.001, 0.01] 
 'reg_dist': [0.0001, 0.001, 0.01] |
| DAN | 'reg': [0.00001, 0.0001, 0.001, 0.01, 0.1, 1.0, 10.0, 100.0, 1000.0] |
| MCC | 'reg': [0.01, 0.1, 1] 
 'temperature': [1, 2, 3] |
| MDD | 'reg': [0.001, 0.01, 0.1] 
 'gamma': [1, 3] |
| SPA | 'reg': [0.001, 0.01, 0.1, 1] 
 'reg_nap': [0, 1] |

