# OpenReview forum: "SKADA-Bench: Benchmarking Unsupervised Domain Adaptation Methods with Realistic Validation On Diverse Modalities"
_TMLR — Accepted by TMLR_

### Review · Reviewer_N4gg · 2025-02-21

**Summary Of Contributions:**

This paper describes a benchmark for domain adaptation (DA) algorithms that utilizes classification datasets from 4 different domains (image, text, time-series and tabular data). It focuses on a fair evaluation of algorithms with strict model selection protocols ensuring that target domain data is not utilized. Algorithms are tuned using a range of unsupervised model selection strategies (scorers), and the performance of scorers is also compared. While the paper focuses primarily on shallow DA algorithms, evaluating 20 such algorithms, it also includes 3 deep DA algorithms in a separate evaluation on a subset of the datasets. No single one DA algorithm was found to outperform others across the range of datasets and the paper attempts to provide guidance for choosing a DA algorithm and scorer in practical situations.

**Audience:**

Yes

**Claims And Evidence:**

Yes

**Requested Changes:**

Suggestions:
- For the correlation metric used for scorers (e.g. Fig 3), Spearman seems more suitable as model selection depends only on the ranking of scores.
- In Sec 4.2 "Of the five methods under consideration" -> should it be six methods?

**Strengths And Weaknesses:**

Strengths:
- The focus on rigorous evaluation with proper model selection respecting the unsupervised DA setting is a key strength of this work.
- A broad range of shallow DA algorithms is evaluated across a range of domains, in combination with 6 different scorers, which is a significant effort.
- The comparison of scorers is in itself useful, though somewhat tangential to the main focus of the work.
- Code for the benchmarking framework will be released upon publication, which will allow others to build on the work.

Weaknesses;
- It is unclear whether the comparisons of DA algorithms are practically relevant, as there are domain-specific DA methods developed for images and text (e.g. the deep DA methods compared in the paper), and moreover many of the methods being compared are not that recent. The most recent shallow DA method is from 2020 and most recent Deep DA method is from 2018.
- Datasets used are also slighly dated. For example, in the image domain, while Office-31 and Office-Home are still widely used for evaluation, the field is moving on to more complex datasets like VisDA, DomainNet and WILDS.
- Recent papers proposing new model selection strategies also include comparisons of various scorers, though using deep DA methods.

Overall, the paper is to be commended for its rigorous approach and systematic evaluation. However, the methods compared are rather old so the practical relevance of the work is somewhat diminished. In addition, a key message that unsupervised model selection should be used for realistic evaluation purposes has already been communicated in several previous works, as indicated in the paper. To be more relevant the paper should focus on settings/datasets where shallow DA methods are state-of-the-art, deep DA methods, or a specific domain, and perform a deeper analysis. That said, the comparison of scorers is interesting. By TMLR's criteria, the evidence provided does support the main claims of providing a benchmarking platform and evaluation of a range of methods on several datasets, and the work may be interesting to some in the community.

---

> ### Author Response · Authors · 2025-04-09
>
> We thank the reviewer N4gg for their valuable feedback and for acknowledging the experimental rigor and broad applicability of our benchmark along with its reproducibility. We address the reviewers concerns point by point below.
>
> **Summary of contributions**
> > it also includes 3 deep DA algorithms in a separate evaluation on a subset of the datasets
>
> We acknowledge that the 'Compared Methods' section may have created confusion, as it does not explicitly list all deep DA methods included in our benchmark. To clarify, our study includes **7 deep learning approaches**, with the **most recent one from 2023**.
>
> **Weaknessess**
>
> >1. It is unclear whether the comparisons of DA algorithms are practically relevant, as there are domain-specific DA methods developed for images and text (e.g. the deep DA methods compared in the paper), and moreover many of the methods being compared are not that recent. The most recent shallow DA method is from 2020 and most recent Deep DA method is from 2018.
>
> To clarify, all the methods implemented in our benchmark are not tailored to a specific modality. Indeed, we intentionally avoided highly specialized DA methods to ensure a **broader and more generalizable evaluation**. Recent DA research has largely focused on deep learning, which is why the most recent shallow method is from 2020, but we also implemented SPA which is a deep DA method from 2023. More importantly, we want to highlight that, despite the research community shifting away from **shallow DA**, these methods **remain highly relevant in certain domains, such as biosignals**, where data availability is often limited.
>
> >2. Datasets used are also slighly dated. For example, in the image domain, while Office-31 and Office-Home are still widely used for evaluation, the field is moving on to more complex datasets like VisDA, DomainNet and WILDS.
>
>
> We fully acknowledge that the datasets used in our study are not the ones most commonly employed in recent papers. However, our primary objective was to establish a reproducible and realistic benchmark where new datasets or methods could be added easily. The datasets we selected are those predominantly used in the majority of the compared methods, ensuring a fair and meaningful evaluation.
> By prioritizing reproducibility and accessibility, our work aims **to provide a robust foundation for future research while making it easier for others to extend and build upon our findings**.
>
> >3. Recent papers proposing new model selection strategies also include comparisons of various scorers, though using deep DA methods.
>
> Some studies have proposed new model selection strategies, such as MixVal, but these approaches primarily focus on computer vision and deep learning. In contrast, **our benchmark evaluates multiple scoring methods across different settings**, highlighting the inherent difficulty of **identifying a universal scorer that performs well across diverse domains and modalities**.
>
> > To be more relevant the paper should focus on settings/datasets where shallow DA methods are state-of-the-art, deep DA methods, or a specific domain, and perform a deeper analysis.
>
> We thank the reviewer for the suggestion and we highlight that our main objective was to benchmark shallow methods. The deep learning part primarily serves as an illustration of the limitations of deep learning approaches and to highlight the potential advantages of shallow methods, particularly on small datasets beyond the computer vision domain that remain important in practical applications as illustrated by the very large and stable user base of scikit-learn. Moreover, another of our focus was to benchmark realistic scorers which are rarely considered in the literature or domain specifics. As acknowledge by the reviewer, we believe our benchmark could be valuable to the community with a reproducible implementation and realistic evaluation of DA methods across modalities.
>
> **Requested Changes**
>
> > 1. For the correlation metric used for scorers (e.g. Fig 3), Spearman seems more suitable as model selection depends only on the ranking of scores.
>
> We thank the reviewer for the suggestion and **we added a plot with spearman correlation in Figure 6** in the appendix for the revised version. The Spearman correlation analysis leads to the same conclusion: **Circular Validation shows the highest correlation with the final performance**, further emphasizing its superiority over other scoring methods.
>
> > 2. In Sec 4.2 "Of the five methods under consideration" -> should it be six methods?
>
> We thank the reviewer for pointing out the typo which we have corrected in the revised version.

---

### Review · Reviewer_a9qk · 2025-02-23

**Summary Of Contributions:**

The paper introduces DA-Bench, a comprehensive and reproducible benchmark for evaluating unsupervised domain adaptation methods, addressing gap in assessment across diverse data modalities. The authors evaluate 20 shallow DA methods (reweighting, mapping, subspace alignment, and others) and 3 deep DA methods. They extend beyond the vision-focused benchmarks, offering new insights into DA performance across modalities.

They implement hyperparameter tuning approach, using nested cross-validation with five unsupervised scorers (e.g., Importance Weighted, Circular Validation) to mimic real-world conditions where target labels are unavailable. The benchmark reveals that unsupervised scorers like IW and CircV correlate better and are most effective than others.

The results demonstrate that simple linear methods (e.g., CORAL, LinOT) consistently outperform across multiple datasets offering modest gains, while reweighting suits specific shifts and mapping excels with moderate class counts. Deep DA methods perform well in vision tasks, but shallow methods remain competitive in tabular and biomedical data. By open-sourcing the framework, the authors enable future extensions, making DA-Bench a valuable tool for advancing DA research and application.

The paper claims to compare UDA methods under realistic conditions, addressing critical issues in model selection, performance validation, and benchmarking disclosing the restrictions of current validation strategies, and offering actionable insights for practitioners.

**Audience:**

Yes

**Broader Impact Concerns:**

Can the authors add a broader impact statement section

**Claims And Evidence:**

Yes

**Requested Changes:**

While the paper evaluates three deep DA methods, it does not cover more recent or advanced deep DA techniques (e.g., contrastive learning-based DA, transformer-based DA, or self-supervised DA approaches). This feels narrow compared to the 20 shallow methods across all datasets, potentially underrepresenting deep DA’s capabilities. Expand the Evaluation by adding more state-of-the-art deep DA methods, especially Transformer-based DA models, Contrastive learning-based DA approaches and Self-supervised DA techniques. Also try to test them on across modalities to strengthen the paper’s claim of comprehensive benchmarking.

The timeout for nested loops and single outer split for deep methods shows biased results. Add a discussion to show how these affect performance. Provide computational cost analysis for different DA methods and discuss potential optimizations or alternatives for large-scale settings.

The paper acknowledges that DA methods are highly sensitive to hyperparameters, but does not provide practical solutions. Discuss possible automated hyperparameter tuning approaches and provide best practices for tuning DA methods in real-world scenarios.

The benchmark currently focuses only on classification tasks, which limits its applicability to other areas. Include an evaluation of DA methods for regression tasks, especially for applications like biomedical predictions and financial forecasting.

While DA-Bench covers multiple data modalities, the selection is still limited. Add more diverse datasets, such as satellite imagery, time-series financial data, or speech/audio data.

While pre-processing is mentioned, details are referred to an appendix. More in-text explanation of how pre-processing affects results could improve transparency and interpretation.

**Strengths And Weaknesses:**

Strengths:
The paper evaluates unsupervised domain adaptation methods across diverse modalities (computer vision, NLP, tabular, biomedical, assessing 20 shallow methods across different categories alongside 3 deep DA methods.

The use of nested cross-validation with unsupervised scorers (e.g., IW, CircV) to tune hyperparameters reinforce that DA methods are evaluated fairly in realistic unsupervised settings. It mirrors real-world constraints where target labels are absent, providing a practical benchmark.

The benchopt framework makes it reproducible allowing other researchers to extend it with new DA methods and datasets.

Weaknesses:
While the paper evaluates three deep DA methods, it does not cover more recent or advanced deep DA techniques (e.g., contrastive learning-based DA, transformer-based DA, or self-supervised DA approaches). This feels narrow compared to the 20 shallow methods across all datasets, potentially underrepresenting deep DA’s capabilities.

The timeout (4 hours) for nested loops and single outer split for deep methods might skew results, favoring faster methods or missing optimal deep DA performance. The authors could discuss this trade-off’s impact more explicitly.

The nested cross-validation strategy can be computationally expensive, requiring significant resources. Providing guidance on optimizing computation efficiency or developing faster heuristics for model selection could improve usability

---

> ### Author Response · Authors · 2025-04-09
>
> We thank reviewer a9qk for their valuable feedback and for recognizing the strengths of our benchmark in enabling fair, reproducible evaluations across modalities. We address each point below.
>
> **Weaknesses**
>
> 1. We acknowledge that the 'Compared Methods' section may have caused confusion by not explicitly listing all deep DA methods. To clarify, our study includes 7 deep DA methods, with the most recent from 2023. This is now clearly stated in the revised manuscript.
> However, our benchmark primarily focuses on shallow methods, which are underrepresented in prior work but remain important in practical scenarios—especially on non-vision modalities or small datasets where deep methods may be infeasible. Additionally, many recent deep DA approaches target settings like multi-source or test-time adaptation, which are beyond the scope of our study (pairwise unsupervised DA). We see our work as a stepping stone toward broader evaluations of such emerging techniques.
>
> 2. The 4-hour timeout ensures tractable runtimes across all methods. We report NA only when a method exceeds this limit, which occurred in just two cases (aside from DASVM, which is binary-only), minimizing impact on results (see Table 2). We used a single outer split for deep methods to keep evaluations feasible, reflecting the practical challenge of nested CV in low-data regimes. We now clarify this in the revision.
>
> 3. While nested cross-validation is computationally demanding, it remains the gold standard for unsupervised model selection, especially for small datasets. For shallow methods, this rigor ensures realistic comparisons. That said, our benchmark is fully reproducible, and all results are available, so users need not rerun experiments. Most shallow methods run quickly on a laptop, and all methods complete within four hours, showcasing the practicality of shallow DA. Deep methods, in contrast, will benefit from more scalable evaluations with large-scale datasets—an avenue for future work.
>
> **Requested Changes**
>
> 1. As noted, we include 7 deep DA methods, including recent ones like SPA (2023). These span adversarial (DANN, MDD), OT-based (DeepJDOT), and spectral (SPA) approaches. The benchmark focuses on pairwise unsupervised DA with fair evaluation across medium size datasets and diverse modalities. Incorporating contrastive or transformer-based methods would significantly expand the scope and diverge from our current objectives.
>
> 2. We appreciate the comment on computational cost. Figure 11 in the Appendix shows training/testing time for each method, and we now discuss this in Section 4.1. Timeout had no impact on Table 3 (since there is no "NA"), but we agree that using a single outer split for deep DA may introduce variance. We’ve added a “Limitations and future work” section (4.3), which addresses the need for more scalable deep DA evaluation strategies.
>
> 3. Our benchmark emphasizes the challenges of unsupervised hyperparameter selection under distribution shift. While automated tuning in such settings would be highly valuable, it’s a complex problem requiring separate research. We hope this benchmark lays the groundwork for such efforts.
>
> 4. We chose to focus on classification tasks for consistency. Most DA methods and unsupervised scorers are specifically built for classification, and adapting them to regression would require substantial changes. Our framework is modular and extensible, and we hope it can serve as a foundation for the community to build on. Adding support for regression would be a natural next step, and we encourage future work in this direction.
>
> 5. The benchmark already includes four classification modalities, which is, to our knowledge, unprecedented. Our aim was to make the benchmark reproducible and extensible, rather than exhaustive. Adding tasks like satellite imagery would require significant effort from non-experts, but our framework and code guidelines are designed to support such contributions from the community.
>
> 6. Thank you for this helpful suggestion. We have added a detailed discussion of pre-processing steps in Section 3.3, including how features are extracted for each modality and how dimensionality reduction (e.g., PCA) is applied. These steps ensure compatibility with shallow methods and help manage computational cost—especially for high-dimensional data such as images or text.

---

> > ### Comment · Reviewer_a9qk · 2025-04-10
> > **Official recommendation for the submission**
> >
> > I appreciate the effort the authors put into responding to my feedback in detail. I am satisfied that all concerns have been addressed, and I recommend acceptance

---

### Review · Reviewer_jw5k · 2025-03-26

**Summary Of Contributions:**

This paper introduces DA-Bench, a benchmarking framework aimed at evaluating unsupervised domain adaptation (DA) methods across different data modalities, including computer vision, natural language processing, biomedical, and tabular data. The study addresses the challenge of realistic model selection in DA settings, where hyperparameter tuning is difficult due to the absence of labeled target data. This open-source and extensible work uses nested cross-validation and unsupervised scorers to fairly compare methods.

**Audience:**

Yes

**Broader Impact Concerns:**

Nil.

**Claims And Evidence:**

Yes

**Requested Changes:**

Please go through the Weakness section to find the detail suggestions and changes. In brief, the suggested changes are as follows:

1. The benchmark should include recent state-of-the-art approaches towards optimal model selection.
2. Modify Figure 1 to enhance clarity and add more explanations.
3. The inferences drawn in Section 4.1 need more quantitative analysis mapping their discussions to the empirical results obtained.
4. The paper should include more theoretical analysis and novel ideas towards overcoming the challenges portrayed in the benchmark.

**Strengths And Weaknesses:**

Strength:

1. The study benchmarks DA methods across multiple modalities (CV, NLP, tabular, biomedical), ensuring broad applicability.

2. DA-Bench incorporates unsupervised model selection, making results more practical for real-world applications.

3. The paper also provides practical takeaways on which DA methods perform best under different types of shifts.

Weakness:

1. Although the benchmark evaluates 20 shallow DA models, its scope for deep DA models is very limited. The study evaluates only three deep DA methods, which might not capture the full spectrum of state-of-the-art deep DA approaches. Recent works have progressed the state-of-the-art significantly and is needed to be included in the new benchmark. One such example is compared to DANN (2016), AnyDA [1] has shown to be significantly superior. Other examples of new models can be [2] and [3]. Inclusion of such new works is vital in the benchmark for guiding the community towards optimal model selection.

2. The Figure 1 of the paper needs more clarity and proper explanation. It is difficult to understand the different types of shifts. The authors should add quantitative explanations and discuss more thoroughly what the plots try to convey. The coloring scheme of the graph also needs to be clarified.

3. In Table 2 and Figure 3, the study evaluates several scorers, but most of them show low correlation with actual model performance and have high variance, leading to unreliable hyperparameter selection. This scorer performance variability is a major limitation of the paper because it suggests that DA methods, even when well-designed, may fail in practical deployment due to unreliable validation techniques. The study does a great job of exposing this issue but does not provide a definitive solution. Even considering the "Take-away for DA users" in Section 4.1, the authors should support their discussions with quantitative values and highlight the different empirical results which lead to such inferences.

4. The paper lacks in providing much theoretical analysis regarding why certain DA methods perform better in different shift scenarios. Given that the study identifies major limitations in existing scorers (e.g., poor correlation with true model performance, high variance), it would have been valuable to suggest a new scoring mechanism or perhaps a novel DA scheme backed by theoretical justification to handle the limitations.

[1] Chakraborty, O., Sahoo, A., Panda, R. and Das, A., 2023. AnyDA: Anytime domain adaptation. In The Eleventh International Conference on Learning Representations.

[2] SDAT Rangwani, H.; Aithal, S. K.; Mishra, M.; Jain, A.; and Radhakrishnan, V. B. 2022. A closer look at smoothness in domain adversarial training. In International Conference on Machine Learning, 18378–18399. PMLR.

[3] Zhu, D.; Li, Y.; Shao, Y.; Hao, J.; Wu, F.; Kuang, K.; Xiao, J.; and Wu, C. 2023a. Generalized Universal Domain Adaptation with Generative Flow Networks. arXiv preprint arXiv:2305.04466.

---

> ### Author Response · Authors · 2025-04-09
>
> We thank the reviewer  jw5k for their valuable feedback and for acknowledging the broad applicability and practical benefits of our benchmark across modalities. We address the reviewers concerns point by point below.
>
> **Weaknesses**
> 1. We appreciate the reviewer’s recommendations for additional papers. We acknowledge that the 'Compared Methods' section may have created confusion, as it does not explicitly list all deep DA methods included in our benchmark. To clarify, our study includes **7 deep learning approaches**, with the **most recent one from 2023**. However, our primary focus is on pairwise adaptation, whereas recent deep learning DA works often emphasize multi-source and test-time adaptation.
> Additionally, we would like to stress that our main objective was to benchmark shallow methods. The deep learning benchmark, while included, primarily serves to illustrate the limitations of deep learning approaches and to highlight the potential advantages of shallow methods, particularly on small datasets beyond the computer vision domain that remain important in practical applications as illustrated by the very large and stable user base of scikit-learn.
>
> 2. We thank the reviewer for pointing out the lack of clarity of the figure 1 and for their suggestions which helped us improve it in the revised version of the paper with more details about the shifts assumptions.
>
> 3. We agree with the reviewer that **hyperparameter selection remains one of the main challenges** in unsupervised domain adaptation, and we appreciate the emphasis on this point. Our benchmark highlights that many unsupervised scorers suffer from weak correlation with target accuracy and high variance, as shown in Figure 3. This is an important result because it highlights the remaining challenges in the DA community. Among them, **circular validation clearly stands out** as the most reliable option for shallow methods despite being one of the earliest one proposed in the literature. It achieves the highest Pearson correlation (ρ = 0.71) between the cross-validation score and final target performance, making it a valuable, though still imperfect, proxy. However, its computational cost —requiring both source-to-target and target-to-source training— limits its applicability to deep learning methods.
> Although we do not claim to solve model selection in DA, which is a challenging task, our goal is to expose this limitation clearly and provide a practical benchmark for future research.
>
> 4. We agree with the reviewer on the importance of principled methods for hyperparameter selection. While our paper is mainly empirical, we believe that **carefully evaluating existing scorers** across many modalities and shift types **is a necessary first step** before proposing new ones. The key takeaway is that **most scoring methods show weak or inconsistent correlation with final performance**, making hyperparameter tuning difficult in practice. **Circular validation stands out** as the most reliable scorer for shallow methods, showing the highest correlation with target accuracy. The gap we observe between model performance and scorer reliability highlights the need for better, theory-driven validation techniques. We see DA-Bench as a solid foundation for future work in that direction.
>
>
> **Requested Changes**
>
> > 1. The benchmark should include recent state-of-the-art approaches towards optimal model selection.
>
> We note that the benchmark already includes **MixVal scorer**, a model selection method from **2023** and that our reproducible implementation allows for an easy extension in the future when more scoring method become available.
>
> > 2. Modify Figure 1 to enhance clarity and add more explanations.
>
> We thank the reviewer for the suggestions and have addressed that point by including **a revised version of Figure 1** with improved visual clarity and a more detailed caption explaining each type of shift.
>
> > 3. The inferences drawn in Section 4.1 need more quantitative analysis mapping their discussions to the empirical results obtained.
>
> This section was indeed lacking of quantitative analysis, we thank the reviewer for pointing it out. We added a more detailed analysis in Section 4.1 in the revised version.
>
> > 4. The paper should include more theoretical analysis and novel ideas towards overcoming the challenges portrayed in the benchmark.
>
> We believe such an analysis is beyond the scope of the current benchmark paper. While we include a discussion of which methods perform best under known shifts (using simulated data), developing new theoretical models or scoring methods would require a separate contribution. Our focus is to expose current limitations and provide a reproducible foundation for future work in this direction.

---

### Decision · Action_Editor_KFyF · 2025-06-26

**Recommendation:** Accept as is

**Audience:**

Yes

**Audience Explanation:**

This study systematically evaluates shallow and deep DA methods across diverse modalities such as CV, NLP, table data, and biomedical data, and further addresses the practical scenario of unsupervised model selection. Such practical insights and reproducible benchmark designs are valuable not only for algorithm developers but also for a wide range of researchers seeking to apply machine learning techniques, and are expected to attract significant attention.In particular, the clear contribution to the community through the public release of code is likely to appeal to both academic and practical readers.

**Claims And Evidence:**

Yes

**Claims Explanation:**

The authors evaluated 20 shallow DA methods and 7 deep methods across four modalities, including computer vision, natural language processing, table data, and biomedical data, demonstrating diversity and comprehensiveness in their target applications. Additionally, the comparison of scorers under unsupervised settings and evaluations using nested cross-validation aligned with realistic settings provide insights relevant to real-world applications. Furthermore, the reproducible framework based on benchopt ensures scalability for other researchers, and the consistency between claims and evidence is deemed high.